# A Group Variable Importance Framework for Bayesian Neural Networks

## Abstract

While the success of neural networks has been well-established across a variety of domains, our ability to interpret these methods is still limited. Traditional variable importance approaches in machine learning overcome this issue by providing local explanations about particular predictive decisions — that is, they detail how important any given feature is to the classification of a particular sample in the dataset. However, univariate mapping approaches have been shown across many applications in the literature to generate false positives and negatives in high-dimensional and collinear data settings. In this paper, we focus on the slightly different task of global interpretability where our goal is to identify important groups of variables by aggregating over collections of univariate signals to improve power and mitigate false discovery. In the context of neural networks, a feature is rarely important on its own, so our strategy is specifically designed to leverage partial covariance structures and incorporate variable interactions into our proposed group feature ranking. Here, we extend the recently proposed "RelATive cEntrality" (RATE) measure to the Bayesian deep learning setting. We refer to this approach as the "GroupRATE" criterion. Given a trained network, GroupRATE applies an information theoretic metric to the joint posterior distribution of effect sizes to assess group-level significance of features. Importantly, unlike competing approaches, our method does not require tuning parameters which can be costly and difficult to select. We demonstrate the utility of our framework on both simulated and real data.

## 1 Introduction

Due to their high predictive performance, feedforward neural networks have become increasingly ubiquitous in many fields including computer vision and natural language processing (LeCun et al., 2015). Unfortunately, neural networks operate as "black boxes": users are rarely able to understand the internal workings of the network. As a result, these approaches have not been widely adopted in scientific settings where variable selection tasks are often as important as prediction — one particular example being the identification of biomarkers related to the progression of a disease. While neural networks are beginning to be used in high-risk decision-making fields (e.g., automated medical diagnostics or self-driving cars such as in Lundervold & Lundervold (2019); Rahman et al. (2019)), it is critically important that methods do not make predictions based on artifacts or biases in the training data. Therefore, there is both a strong theoretical and practical motivation to increase the global interpretability of neural networks and to better characterize the types of relationships upon which they rely.

The increasingly important concept of interpretability in machine learning still lacks a well-established definition in the literature. Despite recent surveys (Guidotti et al., 2018; Carvalho et al., 2019; Marcinkevics & Vogt, 2020) and proposed guidelines (Rudin, 2019; Hall, 2019) to address this issue, conflicting views on how interpretability should be evaluated still remain. Variable importance is one possible approach to achieve global interpretability, where the goal is to rank each input feature based on its contributions to predictive accuracy. This is in contrast to local interpretability, which aims to simply provide an explanation behind a specific prediction or group of predictions (Arya et al., 2019; Clough et al., 2019). In this paper, we follow a definition which refers to interpretability as "the ability to explain or to present in understandable terms to a human" (Doshi-Velez & Kim, 2017). To this end, our main contribution is focused on global interpretability: we address the problem of identifying important predictor variables given a trained neural network, focusing especially on settings in which variables (or groups of variables) are intrinsically meaningful.

Univariate mapping approaches have been shown to be underpowered and prone to high false discovery rates in settings when there is complex correlation structure between features or when data is generated by many variables with small effects (Galesloot et al., 2014). As a result, a recent strategy to improve the power of variable importance methods is to leverage the naturally groupings of features that are well defined in different applications. For example, in biomedicine, single-nucleotide polymorphisms (SNPs) fall within the regulatory regions of genes, collections of microbiota form taxa, and medical images contain spatial regions of pixels corresponding to anatomically relevant features. Grouping variables can also dramatically reduce the number of objects to be compared by reducing the resolution at which a system is studied. This leads to a reduction in both the computational cost and statistical complexity (e.g., by reducing the number of tests when working in a hypothesis testing framework). Furthermore, groups may be a more natural scale at which to interpret the system of interest. This is the case in brain magnetic resonance imaging (MRI) scanning, where individual voxels have extremely limited meaning but can be grouped into brain regions that are far more relevant and interpretable (Wehenkel et al., 2018). Lastly, grouping variables before calculating importance can also give statistical benefits when features exhibit a high degree of collinearity within a group. This is the case in medical imaging, where there is high spatial correlation between pixels or voxels, as well as in genetic studies where linkage disequilibrium can cause SNPs within a genomic region to be highly collinear.

To that end, we describe an approach to interpret neural networks using a group-level extension of "RelATive cEntrality" (RATE) criterion (Crawford et al., 2019), a recently-proposed univariate approach for assessing variable importance in Bayesian models. We refer to our new flexible framework as GroupRATE which assesses the importance of groups of variables according to some set of *a priori* annotations or domain knowledge. Most importantly, GroupRATE can be used with any network architecture where some notion of uncertainty can be computed over the predictions. The rest of the paper is structured as follows. Section 2 outlines related work on the interpretation of neural networks. Section 3 describes the univariate RATE computation within the context for which it was originally proposed (Gaussian process regression). Section 4 contains the main methodological innovations of this paper. Here, we present a unified framework under which RATE can be applied to neural networks based on variational Bayes and describe our main innovation GroupRATE for estimating group-level variable importance. In Section 5, we demonstrate the utility of our method in various simulation scenarios and real data applications, and compare our proposed framework to competing approaches. We then close with a discussion.

## 2 Related Work

In the absence of a robustly defined metric for interpretability, most work on neural networks has centered around locally interpretable methods with the goal to explain specific classification decisions with respect to input features (Bach et al., 2015; Ribeiro et al., 2016; Shrikumar et al., 2016; Ancona et al., 2018; Sundararajan et al., 2017; Adebayo et al., 2018). In this work, we focus instead on global interpretability where the goal is to identify predictor variables that best explain the overall performance of a trained model. Previous work in this context have attempted to solve this issue by selecting inputs that maximize the activation of each layer within the network (Erhan et al., 2009). Another viable approach for achieving global interpretability is to train more conventional statistical methods to mimic the predictive behavior of a neural network. This "student" or "mimic" model is then retrospectively used to explain the predictions that a model would make at a global level. Such mimic models are typically trained on the soft labels (the predicted probabilities) output by the network, as these are often more informative than the corresponding hard (class) labels (Ba & Caruana, 2014; Hinton et al., 2015; Che et al., 2016).

For example, using a decision tree (Frosst & Hinton, 2017; Kuttichira et al., 2019) or falling rule list (Wang & Rudin, 2015) can yield straightforward characterizations of predictive outcomes. Unfortunately, these simple models can struggle to mimic the accuracy of neural networks effectively. A random forest (RF) or gradient boosting machine (GBM), on the other hand, is much more capable of matching the predictive power of neural networks. Measures of feature importance can be computed for RFs and GBMs by permuting information within the input variables and examining this null effect on test accuracy, or by calculating Mean Decrease Impurity (MDI) (Breiman, 2001). The ability to establish variable importance in random forests is a significant reason for their popularity in fields such as the life and clinical sciences (Chen et al., 2007), where random forest and gradient boosting machine mimic models have been used as interpretable predictive

models for patient outcomes (Che et al., 2016). A notable drawback of RFs and GBMs is that it can take a significant amount of training time to achieve accuracy comparable to the neural networks that they serve to mimic. This provides motivation for our direct approach, avoiding the need to train a separate model.

## 3 Relevant Background

In this section, we give a brief review on previous results that are relevant to our main methodological innovations for performing group variable importance in Bayesian neural networks. Throughout, we assume that we have access to some trained Bayesian model with the ability to draw samples from its posterior predictive distribution. This reflects the *post-hoc* nature of our objective of finding important subsets of variables.

### 3.1 Effect Size Analogues for Bayesian Nonparametric Models

Assume that we have an $N$-dimensional response vector $\mathbf{y}$ and an $N \times J$ design matrix $\mathbf{X}$ with $N$ observations and $J$ covariates. To begin, we consider a standard linear regression model where

$$\mathbf{y} = \boldsymbol{f} + \boldsymbol{\varepsilon}, \qquad \boldsymbol{f} = \mathbf{X}\boldsymbol{\beta}, \qquad \boldsymbol{\varepsilon} \sim \mathcal{N}\left(\mathbf{0}, \tau^2 \mathbf{I}\right), \tag{1}$$

where $\boldsymbol{\beta}$ is a $J$-dimensional vector of additive effect sizes, $\boldsymbol{\varepsilon}$ is an $N$-dimensional vector of error terms that is assumed to follow a multivariate normal distribution with mean zero $\mathbf{0}$ and scaled variance term $\tau^2$, and $\mathbf{I}$ is an identity matrix. In classical statistics, a least squares estimate of the regression coefficients is defined as the projection of the response variable onto the column space of the data: $\text{Proj}(\mathbf{X}, \mathbf{y}) = \mathbf{X}^\dagger \mathbf{y}$, with $\mathbf{X}^\dagger$ being the Moore-Penrose generalized inverse. For high-dimensional settings with correlated features, one may also consider using regularization via ridge regression such that $\text{Proj}(\mathbf{X}, \mathbf{y}) = (\mathbf{X}^\intercal \mathbf{X} + \vartheta \mathbf{I})^{-1} \mathbf{X}^\intercal \mathbf{y}$ where $\lambda > 0$ is a penalty parameter.

In Bayesian nonparametric models, we relax the additive assumption in the covariates and consider a learned nonlinear function $\boldsymbol{f}$ that has been evaluated on the $N$-observed samples (Kolmogorov & Rozanov, 1960; Schölkopf et al., 2001; 2002)

$$\mathbf{y} = \boldsymbol{f} + \boldsymbol{\varepsilon}, \qquad \boldsymbol{\varepsilon} \sim \mathcal{N}\left(\mathbf{0}, \tau^2 \mathbf{I}\right), \tag{2}$$

where $\boldsymbol{f} = [f(\mathbf{x}_1), \ldots, f(\mathbf{x}_N)]^\intercal$. Previous work considered Gaussian process regression, where $\boldsymbol{f} \sim \mathcal{N}(\mathbf{0}, \mathbf{K})$ lives within a reproducing kernel Hilbert space (RKHS) defined by some nonlinear covariance function $k_{ii'} = k(\mathbf{x}_i, \mathbf{x}_{i'})$ for each element in $\mathbf{K}$ (Rasmussen & Williams, 2006). Here, we consider a mean-field Bayesian neural network constructed such that $\boldsymbol{f}$ is also drawn from a multivariate normal. The effect size analogue, denoted $\widetilde{\boldsymbol{\beta}}$, represents the nonparametric equivalent to coefficient estimates in linear regression using common approaches such as generalized ordinary least squares or regularization. Similarly, this can then be defined as the result of $\text{Proj}(\mathbf{X}, \boldsymbol{f})$ which projects the learned nonlinear vector $\boldsymbol{f}$ onto the original design matrix $\mathbf{X}$ in the following respective ways

$$\widetilde{\boldsymbol{\beta}}_{\text{Linear}} = \mathbf{X}^\dagger \boldsymbol{f}, \qquad \widetilde{\boldsymbol{\beta}}_{\text{Ridge}} = (\mathbf{X}^\intercal \mathbf{X} + \vartheta \mathbf{I})^{-1} \mathbf{X}^\intercal \boldsymbol{f}, \tag{3}$$

with $\vartheta \geq 0$ representing a free regularization parameter. Some intuition for the effect size analogue can be gained as follows. After having fit a probabilistic model, we consider the fitted values $\boldsymbol{f}$ and regress these predictions onto the input variables so as to see how much variance these features explain. This is a simple way of understanding the relationships that the model has learned. The coefficients produced by this linear projection have their normal interpretation: they provide a summary of the relationship between the covariates in $\mathbf{X}$ and $\boldsymbol{f}$. For example, while holding everything else constant, increasing some feature $\mathbf{x}_j$ by 1 will increase $\boldsymbol{f}$ by $\widetilde{\beta}_j$. In the case of kernel machines, theoretical results for identifiability and sparsity conditions of the effect size analogue have been previously developed when using the Moore-Penrose generalized inverse as the projection operator (Crawford et al., 2018).

### 3.2 Univariate Variable Importance using Relative Centrality Measures

Similar to regression coefficients in linear models, effect size analogues are not used to solely determine variable significance. Indeed, there are many approaches to infer univariate associations based on the magnitude of

effect size estimates, but many of these techniques rely on arbitrary thresholding and fail to account for key covarying relationships that exist within the data. The "RelATive cEntrality" measure (or RATE) was developed as a *post-hoc* approach for variable prioritization that mitigates these concerns (Crawford et al., 2019).

Consider a sample from the predictive distribution of $\widetilde{\boldsymbol{\beta}}$, obtained by iteratively transforming draws from the posterior of $\boldsymbol{f}$ via one of the deterministic projections specified in Equation (3). The RATE criterion summarizes how much any one variable contributes to the total information the model has learned. Effectively, this is done by taking the Kullback-Leibler divergence (KLD) between *(i)* the conditional posterior predictive distribution $p(\widetilde{\boldsymbol{\beta}}_{-j} \mid \widetilde{\beta}_j = 0)$ with the effect of the $j$-th predictor being set to zero, and *(ii)* the marginal distribution $p(\widetilde{\boldsymbol{\beta}}_{-j})$ with the effects of the $j$-th predictor being integrated out. In this work, we denote the RATE criterion as the following

$$\gamma_j = \frac{\mathrm{KLD}_j}{\sum_k \mathrm{KLD}_k} \, ,$$

where $\gamma_j$ quantifies the importance of the $j$-th variable in the model and

$$\mathrm{KLD}_j := \mathrm{KL}\left(p(\widetilde{\boldsymbol{\beta}}_{-j}) \,\|\, p(\widetilde{\boldsymbol{\beta}}_{-j} \mid \widetilde{\beta}_j = 0)\right) = \int \log\left(\frac{p(\widetilde{\boldsymbol{\beta}}_{-j})}{p(\widetilde{\boldsymbol{\beta}}_{-j} \mid \widetilde{\beta}_j = 0)}\right) p(\widetilde{\boldsymbol{\beta}}_{-j}) \, \mathrm{d}\widetilde{\boldsymbol{\beta}}_{-j}. \tag{4}$$

Note that the $\mathrm{KLD}_j$ is a non-negative quantity, and equals zero if and only if the $j$-th variable is of little-to-no importance, since removing its effect has no influence on the joint distribution other variable effects. In addition, the RATE criterion is bounded within the range $\gamma_j \in [0, 1]$ and has the natural interpretation of measuring a variable's relative entropy — with a higher value equating to more importance.

### 3.3 Closed-Form Relative Centrality Measures

Under the modeling assumptions for the weight-space Gaussian process in Equation (2), the posterior distribution of the effect size analogue $\widetilde{\boldsymbol{\beta}}$ via the projections specified in Equation (3) is multivariate normal with an empirical mean vector $\boldsymbol{\mu}$ and positive semi-definite covariance/precision matrix $\boldsymbol{\Omega} = \boldsymbol{\Lambda}^{-1}$. Given these values, we may partition conformably for the $j$-th input variable such that

$$\boldsymbol{\mu} = \left( \begin{array}{c} \mu_j \\ \boldsymbol{\mu}_{-j} \end{array} \right), \qquad \boldsymbol{\Omega} = \left( \begin{array}{cc} \omega_j & \boldsymbol{\omega}_{-j}^{\mathsf{T}} \\ \boldsymbol{\omega}_{-j} & \boldsymbol{\Omega}_{-j} \end{array} \right), \qquad \boldsymbol{\Lambda} = \left( \begin{array}{cc} \lambda_j & \boldsymbol{\lambda}_{-j}^{\mathsf{T}} \\ \boldsymbol{\lambda}_{-j} & \boldsymbol{\Lambda}_{-j} \end{array} \right). \tag{5}$$

With these normality assumptions, after conditioning on $\widetilde{\beta}_j = 0$, Equation (4) for the RATE criterion has the following closed-form solution

$$\mathrm{KLD}_j = \frac{1}{2}\left[ \mathrm{tr}(\boldsymbol{\Omega}_{-j}\boldsymbol{\Lambda}_{-j}) - \log|\boldsymbol{\Omega}_{-j}\boldsymbol{\Lambda}_{-j}| - (J-1) + \delta_j \mu_j^2 \right], \tag{6}$$

where $\mathrm{tr}(\cdot)$ is the matrix trace function, and $\delta_j = \boldsymbol{\lambda}_{-j}^{\mathsf{T}}\boldsymbol{\Lambda}_{-j}^{-1}\boldsymbol{\lambda}_{-j}$ characterizes the implied linear rate of change of information when the effect of any predictor is absent — thus, providing a natural (non-negative) numerical summary of the role of each $\widetilde{\beta}_j$ plays in defining the full joint posterior distribution. In other words, $\delta_j$ is larger for variables whose effects also have greater dependency on the effects of other variables. Crawford et al. (2019) show that, in a dataset with a reasonably large number of $J$ features, the term $\mathrm{tr}(\boldsymbol{\Omega}_{-j}\boldsymbol{\Lambda}_{-j}) - \log|\boldsymbol{\Omega}_{-j}\boldsymbol{\Lambda}_{-j}| - (J-1)$ remains relatively equal for each input variable and, thus, makes a negligible contribution to when determining the variable importance. Therefore, in practice, we compute RATE measures using the following approximation

$$\mathrm{KLD}_j \approx \delta_j \mu_j^2 / 2 \, . \tag{7}$$

Note that the scalability of the RATE calculation in Equation (7) (which includes a feature's posterior mean and the joint covariance matrix) is $\mathcal{O}(JN^2 + J^2N + J^4)$ for $N$ observations and $J$ variables. The leading order term is $\mathcal{O}(J^4)$ which is driven by $J$ independent $\mathcal{O}(J^3)$ operations of solving the $(J-1)$-dimensional

linear systems $\delta_j$ for $j = 1, \ldots, J$. This restricts the current implementation of RATE to datasets of size $n \lesssim 10^5$ and $J \lesssim 10^4$ if the system is solved. Fortunately, the matrix $\boldsymbol{\Lambda}_{-j}^{-1}$ differs by only a single row and column between consecutive values of the $j$-th index, meaning that low-rank updates can be used to solve $\delta_j = \boldsymbol{\lambda}_{-j}^{\mathsf{T}} \boldsymbol{\Lambda}_{-j}^{-1} \boldsymbol{\lambda}_{-j}$ in $\mathcal{O}(J^2)$ time using the Sherman-Morrison formula (Hager, 1989). Given the partition in Eq. (5), we can approximate $\boldsymbol{\Lambda}_{-j}^{-1}$ using a rank-1 update for each feature in the model. This can be done by removing the $j$-th row and column from the following matrix

$$\boldsymbol{\Lambda}_{-j}^{-1} \approx \left[ \boldsymbol{\Lambda} - \boldsymbol{\Lambda} \boldsymbol{\omega}_j \boldsymbol{\omega}_j^{\mathsf{T}} \boldsymbol{\Lambda} / \left( 1 + \boldsymbol{\omega}_j^{\mathsf{T}} \boldsymbol{\Lambda} \boldsymbol{\omega}_j \right) \right]_{-j}. \tag{8}$$

Ultimately, this reduces the computational complexity of Equation (7) to just $J$-independent $O(J^2)$ operations which can be parallelized.

### 3.4 Relationship between Relative Centrality and Mutual Information

To build further intuition about centrality measures, we establish a formal connection between the RATE measure and mutual information (MI). By simplifying the definition of mutual information, we have

$$
\begin{aligned}
\mathrm{MI}(\widetilde{\boldsymbol{\beta}}_{-j}, \widetilde{\boldsymbol{\beta}}_j) &= \iint p(\widetilde{\boldsymbol{\beta}}_{-j}, \widetilde{\boldsymbol{\beta}}_j) \log \left( \frac{p(\widetilde{\boldsymbol{\beta}}_{-j}, \widetilde{\boldsymbol{\beta}}_j)}{p(\widetilde{\boldsymbol{\beta}}_{-j}) p(\widetilde{\boldsymbol{\beta}}_j)} \right) \mathrm{d}\widetilde{\boldsymbol{\beta}}_{-j} \, \mathrm{d}\widetilde{\boldsymbol{\beta}}_j \\
&= \iint p(\widetilde{\boldsymbol{\beta}}_j) p(\widetilde{\boldsymbol{\beta}}_{-j} \,|\, \widetilde{\boldsymbol{\beta}}_j) \log \left( \frac{p(\widetilde{\boldsymbol{\beta}}_{-j} \,|\, \widetilde{\boldsymbol{\beta}}_j)}{p(\widetilde{\boldsymbol{\beta}}_{-j})} \right) \mathrm{d}\widetilde{\boldsymbol{\beta}}_{-j} \, \mathrm{d}\widetilde{\boldsymbol{\beta}}_j \\
&= \int p(\widetilde{\boldsymbol{\beta}}_j) \, \mathrm{KL}\left( p(\widetilde{\boldsymbol{\beta}}_{-j} \,|\, \widetilde{\boldsymbol{\beta}}_j) \,\|\, p(\widetilde{\boldsymbol{\beta}}_{-j}) \right) \mathrm{d}\widetilde{\boldsymbol{\beta}}_j \, .
\end{aligned}
\tag{9}
$$

While the RATE criterion compares the marginal distribution $p(\widetilde{\boldsymbol{\beta}}_{-j})$ to the conditional distribution $p(\widetilde{\boldsymbol{\beta}}_{-j} \,|\, \widetilde{\beta}_j = 0)$ with the effect of the $j$-th predictor being set to zero, the mutual information criterion compares $p(\widetilde{\boldsymbol{\beta}}_{-j})$ to the conditional distribution $p(\widetilde{\boldsymbol{\beta}}_{-j} \,|\, \widetilde{\beta}_j)$ for all the possible values of $\widetilde{\beta}_j$. Whenever the effect size analogue follows a normal distribution $\widetilde{\boldsymbol{\beta}} \sim \mathcal{N}(\boldsymbol{\mu}, \boldsymbol{\Omega})$, the unnormalized RATE criterion for the $j$-th variable is given by Equation (6). In the same setting, the mutual information can be computed as

$$\mathrm{MI}(\widetilde{\boldsymbol{\beta}}_{-j}, \widetilde{\boldsymbol{\beta}}_j) = \frac{1}{2} \log \left( \delta_j |\boldsymbol{\Omega}_{-j}| |\boldsymbol{\Omega}|^{-1} \right) , \tag{10}$$

where the above is equal to 0 if and only if $\widetilde{\boldsymbol{\beta}}_{-j}$ and $\widetilde{\boldsymbol{\beta}}_j$ are independent. To see the difference between the two information theoretic measures in Equations (6) and (10), notice that $\mathrm{MI}(\widetilde{\boldsymbol{\beta}}_{-j}, \widetilde{\boldsymbol{\beta}}_j)$ only depends on the values of the covariance/precision matrix $\boldsymbol{\Omega} = \boldsymbol{\Lambda}^{-1}$. This is in contrast to the RATE criterion which also takes the posterior mean (or marginal effect) of input features $\boldsymbol{\mu}$ into account when determining variable importance. Therefore, if a feature is only marginally associated with an outcome but does not have any significant covarying relationships with other variables in the data, RATE will still identify this feature as being an important predictor.

## 4 Variable Importance in Bayesian Neural Networks

We now detail the main methodological contributions of this paper. First, we describe a motivating Bayesian neural network framework which utilizes variational inference. Next, we propose a new effect size analogue projection that is more robust to collinear input data. Lastly, we extend the univariate RATE criterion to also consider group-level variable importance (i.e., assessing the association of collections of features) with an approach that we refer to as the GroupRATE measure. Importantly, this extension also has a closed-form for scalable implementation in high-dimensional settings. Furthermore, to our knowledge, grouped variable importance has not yet been studied for neural networks, despite several analogous works for other supervised models (Yuan & Lin, 2006; Simon et al., 2013; Gregorutti et al., 2015; Wehenkel et al., 2018).

### 4.1 Motivating Neural Network Architecture

In this section, we take a probabilistic view on prediction which is made possible by using a Bayesian neural network. In contrast to a "standard" neural network, which uses maximum likelihood point-estimates for its parameters, a Bayesian neural network assumes a prior distribution over its weights. The posterior probability over the weights, learned during the training phase, can then be used to compute the posterior predictive distribution. Once again, we consider a general predictive task with an $N$-dimensional set of response variables $\mathbf{y}$ and an $N \times J$ design matrix $\mathbf{X}$ with $p$ covariates. For this problem, we assume the following hierarchical network architecture to learn the predicted response in the data

$$\widehat{\mathbf{y}} = \sigma(\boldsymbol{f}), \qquad \boldsymbol{f} = \mathbf{H}(\boldsymbol{\theta})\mathbf{w} + \mathbf{b}, \qquad \mathbf{w} \sim \pi, \tag{11}$$

where $\sigma(\cdot)$ is a link function, $\boldsymbol{\theta}$ is a vector of inner layer weights, and $\boldsymbol{f}$ is an $N$-dimensional vector of smooth latent values or "functions" that need to be estimated. Here, we use $\mathbf{H}(\boldsymbol{\theta})$ to denote an $N \times L$ matrix of activations from the penultimate layer (which are fixed given a set of inputs $\mathbf{X}$ and point estimates for the inner layer weights $\boldsymbol{\theta}$), $\mathbf{w} \sim \pi$ is a $L$-dimensional vector of weights at the output layer assumed to follow prior distribution $\pi$, and $\mathbf{b}$ is an $N$-dimensional vector of the deterministic bias that is produced during the training phase.

The hierarchical structure of Equation (11) is motivated by the fact that we are most interested in the posterior distribution of the latent variables when computing the effect size analogues and, subsequently, interpretable RATE measures. To this end, we may logically split the network architecture into three key components: *(i)* an input layer of the original predictor variables, *(ii)* hidden layers where parameters are deterministically computed, and *(iii)* the outer layer where the parameters and activations are treated as random variables. Since the resulting functions are a linear combination of these components, their joint distribution will be closed-form if the posterior distribution of the weight parameters can also be written in closed-form. Restricting that only the weights in the outer layer are stochastic also brings computational benefits during network training as it drastically reduces the number of parameters (versus learning a posterior for every parameter in the network).

There are two important features that come with this neural network specification. First, we may easily generalize this type of architecture to different predictive tasks through the link function $\sigma(\cdot)$. For example, we may apply our model to the classification problem by increasing the number of output nodes to match the number of categories and redefining link function to be the sigmoid function. Regression is even simpler where we would let the link function be the identity. Second, the structure of the hidden layers can be of any size or type, provided that we have access to draws of the posterior predictive distribution for the response variables. Ultimately, this flexibility means that a wide range of existing probabilistic network architectures can be easily modified to be used with RATE. A simple example of this architecture is illustrated in Figure 1.

### 4.2 Posterior Inference with Variational Bayes

As the size of datasets in many application areas continues to grow, it has become less feasible to implement traditional Markov Chain Monte Carlo (MCMC) algorithms for inference. This has motivated approaches for supervised learning that are based on variational Bayes and the stochastic optimization of a variational lower bound (Hinton & Van Camp, 1993; Barber & Bishop, 1998; Graves, 2011). In this work, we use variational Bayes because it has the additional benefit of providing closed-form expressions for the posterior distribution of the weights in the outer layer $\mathbf{w}$ and, subsequently, the functions $\boldsymbol{f}$. Here, we first specify a prior $\pi(\mathbf{w})$ over the weights and replace the intractable true posterior $p(\mathbf{w} \mid \mathbf{y}) \propto p(\mathbf{y} \mid \mathbf{w})\pi(\mathbf{w})$ with an approximating family of distributions $q_{\boldsymbol{\phi}}(\mathbf{w})$. The overall goal of variational inference is to select the member of the approximating family that is closest to the true posterior which is done by minimizing the divergence $\mathrm{KL}(q_{\boldsymbol{\phi}}(\mathbf{w}) \,\|\, p(\mathbf{w} \mid \mathbf{y}))$, with respect to variational $\boldsymbol{\phi}$. This is equivalent to maximizing the so-called variational lower bound.

Since the architecture specified in Equation (11) contains point estimates at the hidden layers, we cannot train the network by simply maximizing the lower bound with respect to the variational parameters. Instead, all parameters must be optimized jointly as follows

$$\underset{\boldsymbol{\phi}, \boldsymbol{\theta}}{\arg\max} \; \mathbb{E}_{q_{\boldsymbol{\phi}}(\mathbf{w})} \left[ \log p(\mathbf{y} \mid \mathbf{w}, \boldsymbol{\theta}) \right] - \eta \, \mathrm{KL}(q_{\boldsymbol{\phi}}(\mathbf{w}) \,\|\, \pi(\mathbf{w})) \tag{12}$$

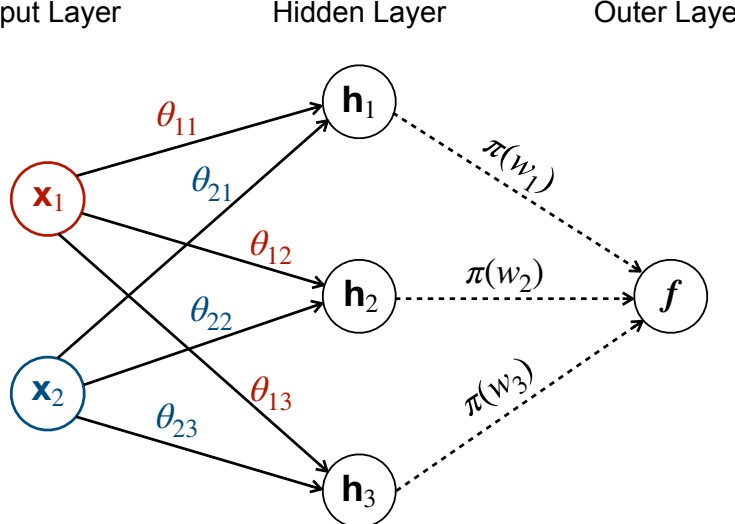

Figure 1: An example of the probabilistic neural network architecture used in this work. The first layer weights $\boldsymbol{\theta}$ are point estimates, while the outer layer weights $\mathbf{w}$ are assumed to be random variables following some prior distribution $\pi = (\pi(w_1), \pi(w_2), \pi(w_3))$. The input variables $\mathbf{x}_1$ and $\mathbf{x}_2$ are fed through the hidden layers $(\mathbf{h}_1, \mathbf{h}_2, \mathbf{h}_3)$ as linear combinations with their corresponding weights $\boldsymbol{\theta}$. Estimates of the predicted functions $\boldsymbol{f}$ are obtained via a linear combination of the activations and samples from the posterior distribution of the outer layer weights $\mathbf{w} = (w_1, w_2, w_3)$. Note that this figure does not include the deterministic bias terms used in Equation (11) for simplicity.

where $\eta \geq 0$ denotes a divergence regularization term (Higgins et al., 2016). We then use stochastic optimization to train the network. Depending on the chosen variational family, the gradients of the minimized $\mathrm{KL}(q_{\phi}(\mathbf{w}) \,\|\, \pi(\mathbf{w}))$ may be available in closed-form, while gradients of the log-likelihood $\log p(\mathbf{y} \,|\, \mathbf{w}, \boldsymbol{\theta})$ are evaluated using Monte Carlo samples and the local reparameterization trick (Kingma et al., 2015). Following this procedure, we obtain an optimal set of parameters for $q_{\phi}(\mathbf{w})$, with which we can sample posterior draws for the outer layer.

In this work, we will use isotropic Gaussians as the family of approximating distributions

$$q_{\phi}(\mathbf{w}) = \mathcal{N}(\mathbf{m}, \mathbf{V}), \quad \phi = \{\mathbf{m}, \mathbf{V}\}, \tag{13}$$

with mean vector $\mathbf{m}$ and a diagonal covariance matrix $\mathbf{V}$. This makes the mean-field assumption that the variational posterior fully factorizes over the elements of $\mathbf{w}$ (Blei et al., 2017). One advantage of this choice is that it ensures that the predicted functions $\boldsymbol{f}$ will follow a multivariate Gaussian distribution as well. Using Equations (11) and (13), we may derive the implied distribution over the latent values using the affine transformation

$$\boldsymbol{f} \,|\, \mathbf{X}, \mathbf{y} \sim \mathcal{N}(\mathbf{H}(\boldsymbol{\theta})\mathbf{m} + \mathbf{b}, \mathbf{H}(\boldsymbol{\theta})\mathbf{V}\mathbf{H}(\boldsymbol{\theta})^{\mathsf{T}}). \tag{14}$$

While the elements of $\mathbf{w}$ are independent, dependencies in the input data (via the hidden activations $\mathbf{H}(\boldsymbol{\theta})$) induce a non-diagonal covariance between the elements of $\boldsymbol{f}$.

### 4.3 Effect Size Analogues via Covariance Projection Operators

After having conducted (variational) Bayesian inference, we now have access to (empirical) draws from the posterior $p(\boldsymbol{f} \,|\, \mathbf{X}, \mathbf{y})$ which we can use to define an effect size analogue for neural networks. In practice, we could use the Moore-Penrose generalized inverse as proposed in Equation (3) but, in the case of highly correlated inputs, these operators can suffer from instability (see a small simulation study in Appendix A), explaining the well-known phenomenon of linear regression suffering in the presence of collinearity. While

regularization poses a viable solution to this problem, the selection of an optimal penalty parameter is not always a straightforward task (see $\vartheta$ in the second half of Equation (3)). As a result, we propose a much simpler projection operator that is particularly effective in application areas where data measurements can be perfectly collinear (e.g., pixels in an image). Our solution is to use a linear measure of dependence separately for each predictor based on the sample covariance. Namely, for each of the $J$ input variables

$$\widetilde{\boldsymbol{\beta}}_{\mathrm{Cov}} = \mathrm{cov}(\mathbf{X}, \boldsymbol{f}) = \mathbf{X}^\intercal \mathbf{C} \boldsymbol{f}/(N-1), \tag{15}$$

where $\mathrm{cov}(\mathbf{X}, \boldsymbol{f}) = [\mathrm{cov}(\mathbf{x}_1, \boldsymbol{f}), \ldots, \mathrm{cov}(\mathbf{x}_J, \boldsymbol{f})]$ is based on the sample covariance, $\mathbf{C} = \mathbf{I} - \mathbf{1}\mathbf{1}^\intercal/N$ denotes a centering matrix, $\mathbf{I}$ is an $N$-dimensional identity matrix, and $\mathbf{1}$ is an $N$-dimensional vector of ones. Probabilistically, since the posterior of the function values $\boldsymbol{f}$ is normally distributed according to Equation (14), the above is equivalent to assuming that $\widetilde{\boldsymbol{\beta}}_{\mathrm{Cov}} \,|\, \mathbf{X}, \mathbf{y} \sim \mathcal{N}(\boldsymbol{\mu}, \boldsymbol{\Omega})$ where

$$\boldsymbol{\mu} = \frac{1}{N-1}\mathbf{X}^\intercal \mathbf{C}\mathbf{H}(\boldsymbol{\theta})\mathbf{m}, \qquad \boldsymbol{\Omega} = \frac{1}{(N-1)^2}\mathbf{X}^\intercal \mathbf{C}\mathbf{H}(\boldsymbol{\theta})\mathbf{V}\mathbf{H}(\boldsymbol{\theta})^\intercal \mathbf{C}^\intercal \mathbf{X}. \tag{16}$$

These moments, along with empirical estimates of the precision matrix $\boldsymbol{\Omega} = \boldsymbol{\Lambda}^{-1}$, can be used directly in Equations (6)-(8) to compute RATE measures for univariate prioritization of each input variable. Intuitively, each element in $\widetilde{\boldsymbol{\beta}}_{\mathrm{Cov}}$ represents some measure of how well the original data at the input layer explains the variation between observations in $\mathbf{y}$. Moreover, under this approach, if two predictors $\mathbf{x}_j$ and $\mathbf{x}_k$ are almost perfectly collinear, then the corresponding effect sizes will also be very similar since $\mathrm{cov}(\mathbf{x}_j, \boldsymbol{f}) \approx \mathrm{cov}(\mathbf{x}_k, \boldsymbol{f})$. To build a better intuition for identifiability under this covariance projection, recall simple linear regression where ordinary least squares (OLS) estimates are unique modulo the span of the data (Wold et al., 1984). A slightly different issue will arise for the effect size analogues computed via Equation (15), where now two estimates are unique modulo the span of a vector of ones, or $span\{\mathbf{1}\}$. We now make the following formal statement.

**Claim 4.1.** *Two effect size analogues computed via the covariance projection operators, $\widetilde{\boldsymbol{\beta}}_1 = cov(\mathbf{X}, \boldsymbol{f}_1)$ and $\widetilde{\boldsymbol{\beta}}_2 = cov(\mathbf{X}, \boldsymbol{f}_2)$, are equivalent if and only if the corresponding functions are related by $\boldsymbol{f}_1 = \boldsymbol{f}_2 + c\mathbf{1}$, where $\mathbf{1}$ is a vector of ones and $c$ is some arbitrary constant.*

The proof of this claim is trivial and follows directly from the covariance being invariant with respect to changes in location. Other proofs connecting this effect size to classic statistical measures can be found in the Appendix B.

### 4.4 Extension of Relative Centrality Measures for Groups of Variables

In many applications, variable selection and prioritization approaches have been shown to be underpowered in settings with small signal-to-noise ratios. For example, in genome-wide association studies, univariate association mapping for single nucleotide polymorphisms (SNPs) can be underpowered for "polygenic" traits which are generated by many mutations of small effect (Manolio et al., 2009; Visscher et al., 2012; Zhou et al., 2013; Yang et al., 2014; Bulik-Sullivan et al., 2015; Wray et al., 2018). To mitigate this issue, recent work have extended methodology to assess the joint global importance for multiple input variables at a time. In the case of genetics, one can use prior knowledge about how groups of SNPs within a particular genomic region are combined (e.g., in a gene or signaling pathway) to detect biologically relevant disease mechanisms underlying complex traits (Liu et al., 2010; Wu et al., 2010; Carbonetto & Stephens, 2013; de Leeuw et al., 2015; Lamparter et al., 2016; Nakka et al., 2016; Zhu & Stephens, 2018; Sun et al., 2019).

The univariate RATE criterion in Equation (6) can also be extended for these types of "set-based" analyses. Assume that we have $G$-predefined annotations $\{\mathcal{A}_1, \ldots, \mathcal{A}_G\}$ which detail how different variables are related to each other. Let each group $g$ represent a known collection of variables $j \in \mathcal{A}_g$ with cardinality $|\mathcal{A}_g|$. As done in the univariate case, once we have access to draws from the posterior distribution of the effect size analogue $\widetilde{\boldsymbol{\beta}}$, we may conformably partition the mean vector and covariance/precision matrices with respect to the $g$-th group of input variables as follows

$$\boldsymbol{\mu} = \begin{pmatrix} \boldsymbol{\mu}_g \\ \boldsymbol{\mu}_{-g} \end{pmatrix}, \qquad \boldsymbol{\Omega} = \begin{pmatrix} \boldsymbol{\Omega}_g & \boldsymbol{\Omega}_{-g}^{*\intercal} \\ \boldsymbol{\Omega}_{-g}^* & \boldsymbol{\Omega}_{-g} \end{pmatrix}, \qquad \boldsymbol{\Lambda} = \begin{pmatrix} \boldsymbol{\Lambda}_g & \boldsymbol{\Lambda}_{-g}^{*\intercal} \\ \boldsymbol{\Lambda}_{-g}^* & \boldsymbol{\Lambda}_{-g} \end{pmatrix}.$$

where $\boldsymbol{\Omega}^*_{-g}$ and $\boldsymbol{\Lambda}^*_{-g}$ are used to denote the covariance and precision matrices between variables inside and outside of the annotated set $\mathcal{A}_g$, respectively. Following the same logic used to derive Equation (6), the RATE criterion to assess the centrality of group $g$ is given as

$$\text{KLD}_g = \frac{1}{2}\left[\text{tr}(\boldsymbol{\Omega}_{-g}\boldsymbol{\Lambda}_{-g}) - \log|\boldsymbol{\Omega}_{-g}\boldsymbol{\Lambda}_{-g}| - (J - |\mathcal{A}_g|) + \boldsymbol{\mu}_g^\mathsf{T}\boldsymbol{\Delta}_g\boldsymbol{\mu}_g\right], \tag{17}$$

where now $\boldsymbol{\Delta}_g = \boldsymbol{\Lambda}^{*\mathsf{T}}_{-g}\boldsymbol{\Lambda}^{-1}_{-g}\boldsymbol{\Lambda}^*_{-g}$ and characterizes the implied linear rate of change of information when the effect of all predictors in the $g$-th group are absent from the model. We refer to the scaled extension of Equation (17) as the GroupRATE criterion. Note that, in practice, the cardinality of groups can differ which may introduce bias in the GroupRATE. To mitigate this bias, we divide the KLD by the size of each group

$$\gamma_g = \frac{\text{KLD}_g/|\mathcal{A}_g|}{\sum_l \text{KLD}_l/|\mathcal{A}_l|} \tag{18}$$

which effectively penalizes the GroupRATE measures for larger groups. This simple correction results in the median correlation between the KL divergences and the size of any group to be zero (see Results).

## 5 Results

In this section, we illustrate the performance of our interpretable Bayesian neural network framework with GroupRATE for prioritizing groups of variables in regression settings. Here, the goal is to show how determining group variable importance for a trained neural network with the RATE measure compares with commonly used group-level modeling techniques in the field. Finally, we examine the potential of our approach in real datasets from genetics and biomedical imaging, respectively.

### 5.1 Simulation Study

For all assessments with synthetic data, we consider a simulation design that is often used to explore the power statistical methods (Crawford et al., 2018; 2019). Once again let $\mathbf{X}$ denote a design matrix of $N$ independent observations with $J$ predictor variables. In this study, we assume that these features are sampled from a zero-mean log-normal distribution such that

$$\log \mathbf{x} \sim \mathcal{N}(\mathbf{0}, \boldsymbol{\Sigma})$$

where $\boldsymbol{\Sigma} = 0.9\boldsymbol{\Sigma}_{\text{grp}} + 0.1\boldsymbol{\Sigma}_{\text{bg}}$ is a combination of group-dependent covariance $\boldsymbol{\Sigma}_{\text{grp}}$ and background covariance $\boldsymbol{\Sigma}_{\text{bg}}$ structures, respectively. Here, we assume that the background covariance follows an inverse-Wishart distribution $\boldsymbol{\Sigma}_{\text{bg}} \sim \mathcal{W}^{-1}(\mathbf{I}, J + 3)$ with an identity scale matrix and $J + 3$ degrees of freedom. Briefly, the inverse-Wishart is a distribution over positive-definite matrices and is a conjugate prior to the covariance of a multivariate Gaussian. The degrees of freedom in the inverse-Wishart controls the concentration of the density around the scale matrix, with larger values increasing this concentration. We assume that the structure of the group covariance $\boldsymbol{\Sigma}_{\text{grp}}$ is block-diagonal with the blocks of non-zero components corresponding to annotated groups $\{\mathcal{A}_1, \dots, \mathcal{A}_G\}$ and zeros everywhere else. Namely, this structure is given is

$$\boldsymbol{\Sigma}_{\text{grp}} = \begin{bmatrix} \boldsymbol{\Sigma}^1_{\text{grp}} & \mathbf{0} & \cdots & \mathbf{0} \\ \mathbf{0} & \boldsymbol{\Sigma}^2_{\text{grp}} & \cdots & \mathbf{0} \\ \vdots & \vdots & \ddots & \vdots \\ \mathbf{0} & \mathbf{0} & \cdots & \boldsymbol{\Sigma}^G_{\text{grp}} \end{bmatrix}$$

where we also allow an inverse-Wishart distribution over the groups $\boldsymbol{\Sigma}^g_{\text{grp}} \sim \mathcal{W}^{-1}(\mathbf{I}, |\mathcal{A}_g| + 3)$. The group structure is an important part of these simulations. In these simulations, we assume that there are $G$ groups with the sizes of each group being randomly determined via a multinomial distribution

$$|\mathcal{A}_1|, \dots, |\mathcal{A}_G| \sim \text{Multinomial}(J, 1/J)$$

which enforces $\sum_g |\mathcal{A}_g| = J$. A single sample of $\boldsymbol{\Sigma}$ is shown in Figure 2. This construction of features ensures there is group-dependent structure in the covariance while also containing non-trivial relationships between other variables.

To complete the simulations, we first randomly select a subset of associated groups and then we use the design matrix $\mathbf{X}$ in the following generative linear model

$$\mathbf{y} = \sum_{c \in \mathcal{C}} \mathbf{x}_c \beta_c + \mathbf{W}\boldsymbol{\theta} + \boldsymbol{\varepsilon}, \qquad \boldsymbol{\varepsilon} \sim \mathcal{N}(\mathbf{0}, \tau^2 \mathbf{I}) \tag{19}$$

where $\mathbf{y}$ is an $N$-dimensional synthetic response vector; $\mathcal{C}$ represents the set of all causal features in the randomly selected causal groups; $\mathbf{x}_c$ is the $c$-th causal feature vector with a corresponding nonzero additive effect size $\beta_c$; $\mathbf{W}$ is an $N \times M$ dimensional matrix which holds all pairwise interactions between the causal features, with the columns of this matrix assumed to be the Hadamard (element-wise) product between feature vectors of the form $\mathbf{x}_j \circ \mathbf{x}_k$ for the $j$-th and $k$-th features; $\boldsymbol{\theta}$ is the $M$-dimensional vector of interaction effect sizes; and $\boldsymbol{\varepsilon}$ is an $N$-dimensional vector of environmental noise. In these simulations, we assume that the total variation of the synthetic response variable is $\mathbb{V}[\mathbf{y}] = 1$. Here, we allow the additive and interaction effect sizes to be randomly drawn from standard normal distributions. Next, we scale the additive, pairwise interactions, and the environmental noise terms so that they collectively explain a fixed proportion of this variance where

$$\mathbb{V}\left[\sum_{c \in \mathcal{C}} \mathbf{x}_c \beta_c\right] = \rho v^2, \qquad \mathbb{V}[\mathbf{W}\boldsymbol{\theta}] = (1 - \rho)v^2, \qquad \mathbb{V}[\boldsymbol{\varepsilon}] = 1 - v^2. \tag{20}$$

Intuitively, $v^2$ determines how much variance in the simulated response is due to signal versus noise, while $\rho$ is a mixture parameter which determines how much of the signal is driven by linear versus nonlinear effects. Given the simulation procedure above, we fix $v^2 = 0.8$ and $J = 10^3$ features. We then simulate a wide range of scenarios by varying the following settings:

- sample size: $N = 10^3$, $2 \times 10^3$, and $5 \times 10^3$ individuals;

- number of associated groups: $G = 50$, $100$, and $200$;

- contribution of additive effects: $\rho = 0.4$ and $0.6$.

In the results below, we will refer to Scenario I as the case where the response is controlled mostly by additivity (i.e., $\rho = 0.75$) and Scenario II as the setting where additivity and pairwise interactions account for an equal share of the signal (i.e., $\rho = 0.5$). All figures and tables show the mean performances (and standard errors) across 50 simulated replicates for each combination of parameter settings.

**Detail of Competing Methods.** The main goal of this simulation study is to compare the performance of our proposed GroupRATE framework to that of other commonly used group-level variable importance methods. To assess the power of GroupRATE, we train a four layer Bayesian neural network with probabilistic weights in the last layer by maximizing the evidence lower bound using the Adam optimizer with a learning rate of $10^{-3}$ for a maximum of 300 epochs (Kingma & Ba, 2014). Training of this model used 80% of the samples while the remaining 20% were held-out as testing data. In addition, we used 10% of the training set as validation data to monitor the behavior of the loss function — where we terminate the training algorithm if the validation loss did not decrease for 30 consecutive epochs (i.e., early stopping). The weight of the divergence regularization term in the evidence lower bound is set to $\eta = 0.3$ throughout (see Equation (12)) and a standard isotropic normal prior is used for all variational parameters. Lastly, rectified linear unit (ReLU) activations are used for hidden layers, each of which contains eight units, and the output layer contains two units and uses an identity activation. Note that no hyper-parameter optimization is performed on the Bayesian neural network as the aim here is not to optimize generalization performance. In a real application, it would be assumed that an extensive hyper-parameter search and cross-validation would have already been performed to obtain a final model. Instead, the task is to interpret this model via a *post-hoc* analysis. We evaluate GroupRATE using the effect size analogue computed with the generalized inverse and

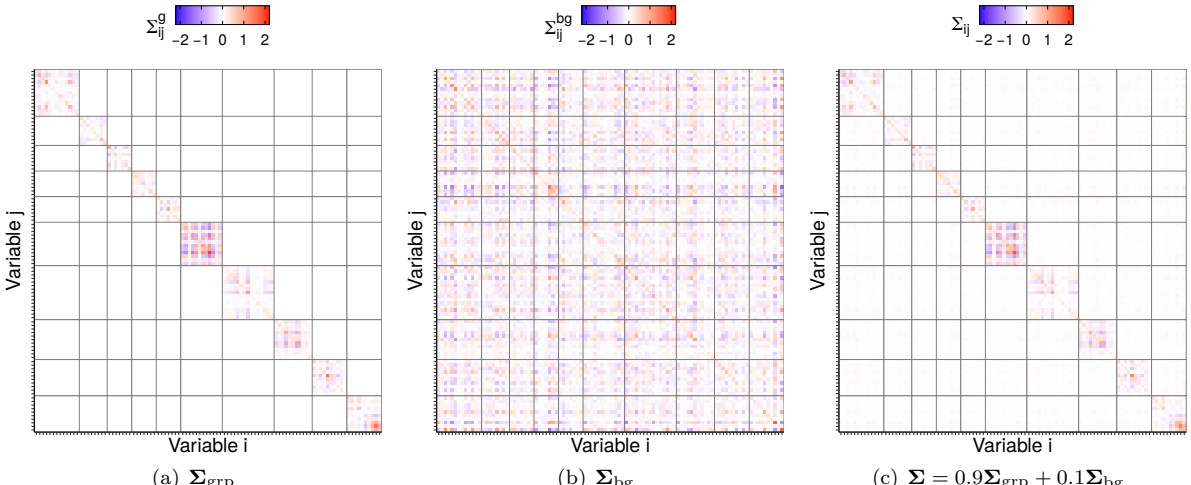

Figure 2: A sample of the covariance structure and variable groupings used in the simulation study. Here, the total covariance $\boldsymbol{\Sigma}$ is a combination of group-dependent covariance $\boldsymbol{\Sigma}_{\mathrm{grp}}$ and background covariance $\boldsymbol{\Sigma}_{\mathrm{bg}}$ structures, respectively. Note that the matrices are partitioned based on the group structure.

ridge regularization projections from Equation (3), as well as with the covariance operator from Equation (15).

We compare the performance of GroupRATE to seven other group-level prioritization approaches which effectively take aggregate summaries over the following univariate variable importance methods:

- vanilla gradients (Simonyan et al., 2014);

- gradients×input (Shrikumar et al., 2016);

- integrated gradients (Sundararajan et al., 2017);

- guided back-propagation (Springenberg et al., 2015);

- smoothed gradients (Smilkov et al., 2017);

- a random forest mimic model with mean decrease Gini variable importance (Breiman, 2001).

The first five of these methods use saliency maps which compute local scores using the gradient of the neural network output with respect to a particular observation in the data. For example, the simplest "vanilla" saliency map attributes the partial derivative $\partial f_i/\partial x_{ij}$ as the importance of the $j$-th feature in the $i$-th sample. In practice, we can then assign global importance using $\sum_{i=1}^{N} |\partial \boldsymbol{f}/\partial \mathbf{x}_j|/N$. In these simulations, we assign group-level importance by simply taking the mean over the univariate global scores for the variables in a given group $\mathcal{A}_g$. For the "vanilla" saliency maps, this done by computing the following

$$s_g = \frac{1}{|\mathcal{A}_g|} \sum_{j \in \mathcal{A}_g} \left[ \frac{1}{N} \sum_{i=1}^{N} \left| \frac{\partial \boldsymbol{f}}{\partial \mathbf{x}_j} \right| \right]. \tag{21}$$

Indeed, the drawbacks of saliency-based methods have been well-documented (Adebayo et al., 2018; Kindermans et al., 2019; Ghorbani et al., 2019), but we include them here due to their popularity. We encourage the reader to see Ancona et al. (2018) for an analysis and comparison of these saliency methods.

Lastly, we consider a random forest mimic model, which is a regression model that takes in the original simulated features $\mathbf{X}$ but is trained on the predicted values $\boldsymbol{f}$ of the fitted Bayesian neural network as response

variables (rather than on the original synthetic outputs $\mathbf{y}$). Using this mimic model approach, we compute global group-level scores by taking the mean-decrease of univariate Gini importance values for each $j$-th variable that have been annotated $g$-th group $\mathcal{A}_g$. Using the mean to aggregate variable-level importances to estimate group-level importance has been investigated by Wehenkel et al. (2018) in the context of 3D brain imaging data applications. Their simulation studies found that using the mean resulted in the best variable selection performance and so that is the same approach we used here.

**Evaluation of Competing Methods.** Figure 3 shows boxplots of the power for each of the different methods over 50 simulated replicates. Here, we assess performance by comparing each method's ability to rank true positives over false positives via the area under their respective receiver operating characteristic curves (AUCs), where a higher value denotes better accuracy in prioritizing the causal groups used in the generative model of the simulations. Overall, there are a few important takeaways from these comparisons. First, most of the methods exhibited better performance as the sample sizes of the simulated data increases (median AUCs $\geq 0.9$). The second key takeaway is that power for all approaches is consistently better when the simulated data are generated by fewer causal groups. The latter occurs because each associated group (and the features assigned to them) make a greater individual contribution to the overall variance for the response (i.e., $\mathbb{V}[\mathbf{y}]/50 > \mathbb{V}[\mathbf{y}]/100 > \mathbb{V}[\mathbf{y}]/200$). Similar trends in performance have been shown during the assessment of high-dimensional variable selection methods in other application areas (Li et al., 2015; Crawford et al., 2017; Zhu & Stephens, 2018; Demetci et al., 2021; Wang et al., 2021; Tang et al., 2022). The one exception that differed from these general trends was the random forest mimic model which had a median AUC $\approx 0.85$ for the simpler tasks (i.e., a combination of sample sizes $N$, few causal groups $G$, and variation driven primarily by additivity $\rho$) and suffered a decrease in AUC towards 0.6 for the hardest tasks. The integrated gradients and gradient×input also suffered for more complicated simulation designs.

The GroupRATE AUCs were consistently competitive with the other best-performing methods (guided back-propagation and smoothed gradients) but there were small differences between the projections used to compute the effect size analogues. Using the ridge and generalized inverse projections both generally led to higher AUCs than the covariance projection which we hypothesize is due to features being simulated with relatively simple block-wise correlation structures (see depiction in Figure 2). The groups in this simulation have a range of sizes which, as previously mentioned in Equation (18), may introduce a bias in GroupRATE scores that are not present in the original RATE calculation. Figure 4A shows that the KL divergence values for a group is positively correlated with the group size, especially for the proposed covariance projection. This correlation decreases towards zero as the sample size of the data $N$ increases, but this bias can be mitigated by dividing the KL divergences by the number of features included in each group (Figure 4B).

**Run Times and Scalability of Competing Methods.** One area in which the different methods studied in this simulation study differ is in their ability to scale to high-dimensional data settings. In this section, we compare this scalability. Since each of these methods perform *post-hoc* variable importance, our evaluation of their respective computational costs does not include the time spent training a model beforehand.

The ten methods that we consider can be divided into three classes based on how they compute group-level importance scores. The first three approaches use GroupRATE with effect size analogues that are computed with different projection operators. The ridge and generalized inverse projections both require a singular value decomposition of the design matrix $X$, which has an $\mathcal{O}(N^2 J)$ running time for $N$ samples and $J$ features. On the other hand, the covariance projection only involves a $\mathcal{O}(N^2 J)$ matrix multiplication, which has the same asymptotic running time as the singular value decomposition but is cheaper in terms of wall clock time. This makes the covariance projection the cheapest computationally of the three effect size analogue projections. Once the effect sizes have been computed, we must empirically compute the moments of their posterior distribution $\boldsymbol{\mu}$, $\boldsymbol{\Omega}$, and $\boldsymbol{\Lambda}$ via an additional $\mathcal{O}(JN^2 + J^2 N)$ matrix multiplication. The final step is solving Equations (17) and (18) for each of the $G$ annotated groups, which requires $G$ independent solutions of a linear system, each of which are $\mathcal{O}(J^3)$. Therefore, the computational complexity of the entire GroupRATE calculation is the following

$$\mathcal{O}(JN^2 + J^2 J + GJ^3),\tag{22}$$

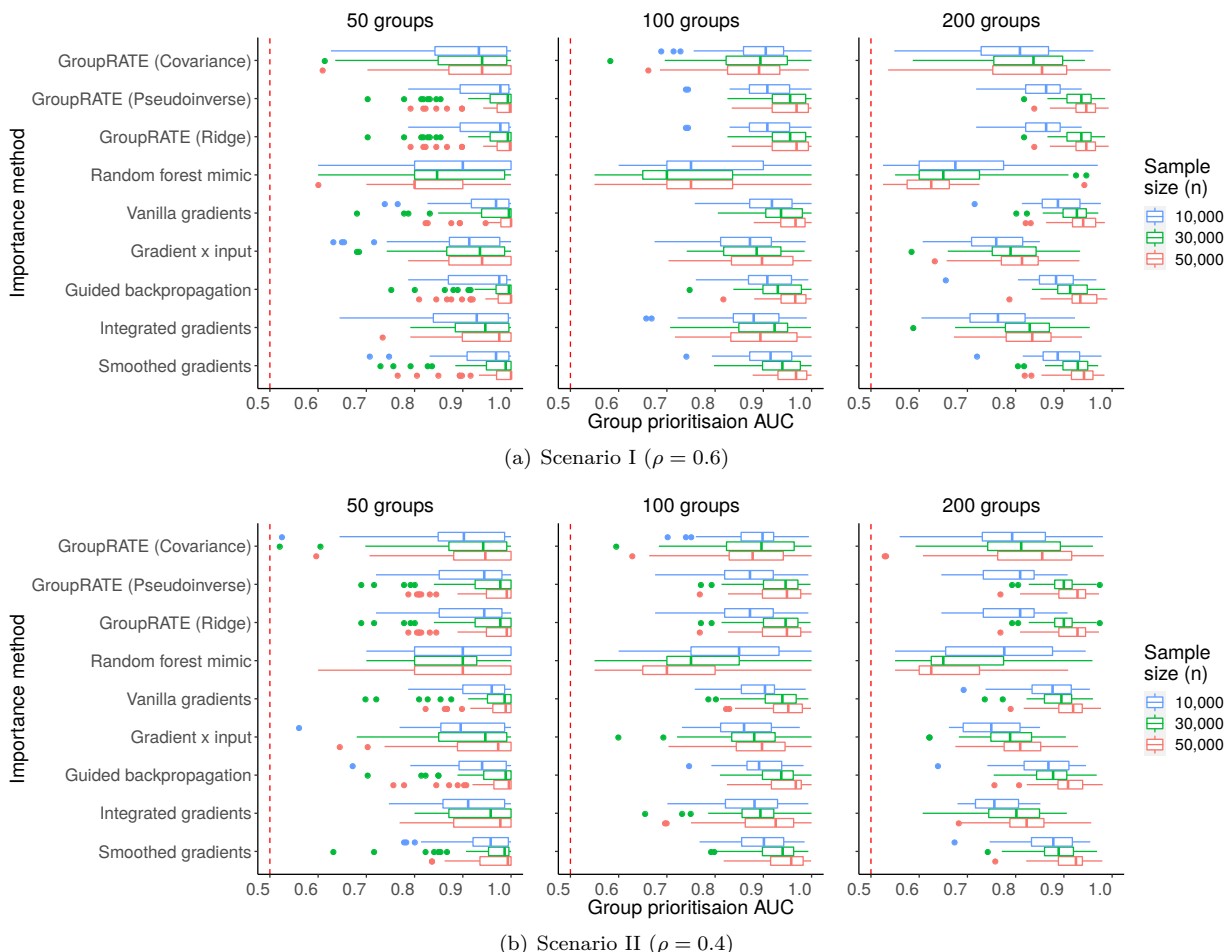

(a) Scenario I ($\rho = 0.6$)

(b) Scenario II ($\rho = 0.4$)

Figure 3: Box plots of the area under the curves (AUCs) for group-level prioritization for GroupRATE and competing variable importance approaches in different simulation scenarios with 50 replicates. The red horizontal line indicates an AUC = 0.5 (i.e., the expected performance of random importance scores). All AUCs are also tabulated in Table S1.

which is dominated by the ESA posterior calculation for $N \gg J$ datasets and by the solution of the KLD for $N \ll J$ datasets. The empirical computation times for the GroupRATE framework with each projection in our simulation study are shown in Figure 5A. These timings are split into the time required to calculate the parameters of the posterior distribution $p(\widetilde{\boldsymbol{\beta}} \,|\, \mathbf{X}, \mathbf{y})$ and the subsequent KL divergence interactions. The covariance projection is the fastest of the three, as expected; however, this difference is on the order of minutes and so is negligible in practice for datasets of these sizes.

The second type of methods are those based on gradients (i.e., saliency maps). Here, these are implemented in `TensorFlow` and so the gradient evaluations are computed efficiently using automatic differentiation. However, within the class of saliency methods, there are those requiring a single gradient evaluation (i.e., vanilla gradients, gradient×input, and guided back-propagation) and those that use repeated evaluations to smooth the gradients (i.e., integrated gradients and smoothed gradients). The random forest mimic model is distinct from the other two types of method as it requires training an entire additional model. This necessitates a hyper-parameter search and cross-validation which, while easy to parallelize, is computationally expensive.

The empirical computation times of the saliency-based methods are shown in Figure 5B. Integrated gradientsand smoothed gradients have the longest running times due to computing repeated gradient evaluations. We want to note that, while these methods are not particularly fast, none of these running times are sufficiently

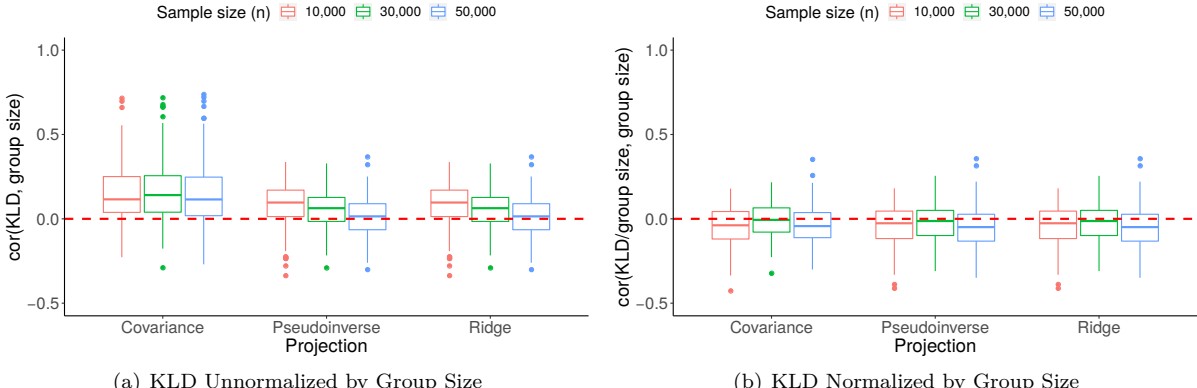

(a) KLD Unnormalized by Group Size        (b) KLD Normalized by Group Size

Figure 4: Panel (a) depicts simulation results showing that the KL divergence values for a group (via Equation (17)) are positively correlated with group size. Panel (b) illustrates that this can be mitigated by dividing each KLD by the corresponding group size when calculating GroupRATE scores.

long to preclude their inclusion in an analysis for datasets of these sizes. The random forest mimic is not plotted here because its cross-validation procedure has a run time that is up to 2 orders of magnitude larger than both GroupRATE and the saliency-based methods.

### 5.2 Assessing Gene Importance in Genome-wide Association Studies

To demonstrate the GroupRATE criterion in real data, we turn to a genome-wide association (GWA) study of a heterogeneous stock of mice dataset from the Wellcome Trust Centre for Human Genetics (Valdar et al., 2006, `http://mtweb.cs.ucl.ac.uk/mus/www/mouse/index.shtml`). We focus on analyzing two quantitative traits: body length and percentage of CD8+ cells. This dataset contains $N \approx 2000$ and $J \approx 10,000$ single nucleotide polymorphisms (SNPs) with minor allele frequencies above 5% — with exact numbers varying slightly depending on the phenotype. In the traditional genome-wide association (GWA) framework, SNPs are individually tested for their marginal importance; however, this approach has been shown to have drawbacks and can suffer from low power when the architecture of a trait is complex (Manolio et al., 2009; Yang et al., 2010; Visscher et al., 2012; Yang et al., 2014). As a result, recent approaches have aimed to combine SNPs within a chromosomal region to detect more biologically relevant genes and enriched pathways (Liu et al., 2010; Ionita-Laza et al., 2013; Nakka et al., 2016; Zhu & Stephens, 2018; Cheng et al., 2019; Demetci et al., 2021). Our interpretable Bayesian neural network framework can be used for similar tasks using GroupRATE. We choose to analyze these particular traits because their architectures represent a realistic mixture of the simulation scenarios we detailed in the previous section. Specifically, these traits have been shown to have various levels of broad-sense heritability (i.e., varying signal-to-noise ratios $v^2$) with different contributions from both additive and non-additive genetic effects (i.e., different values of $\rho$) (Valdar et al., 2006; Chen et al., 2012; Mackay, 2014; Tyler et al., 2016; Crawford et al., 2018; 2019).

Here, we use the Mouse Genome Database (MGD) (Blake et al., 2003, `http://www.informatics.jax.org`) and define groups as collections of SNPs with genomic positions that fall within the same gene (or pseudogene). For simplicity, we eliminate genes with completely overlapping annotations. This resulted in 3,749 total genes (or groups of SNPs) across the 20 chromosomes in the mouse genome to be analyzed. After having trained our neural network, we run GroupRATE on each of these groups using Equation (17) with the three different effect size projection operators to create gene importance scores. We provide summary tables which list all the results after running these three approaches on both the body length and CD8+ phenotypes (Tables S2 and S3). We also use Manhattan plots to visually display the gene-level mapping results across each of these traits, where chromosomes are shown in alternating colors for clarity and notable top ranked genes are highlighted (Figures 6 and S2). Lastly, to further provide contextual relevance of our results, we use the GWAS catalog (`https://www.ebi.ac.uk/gwas/`) to identify molecular categories with an overrepresentation of the most important genes reported by GroupRATE within each trait.

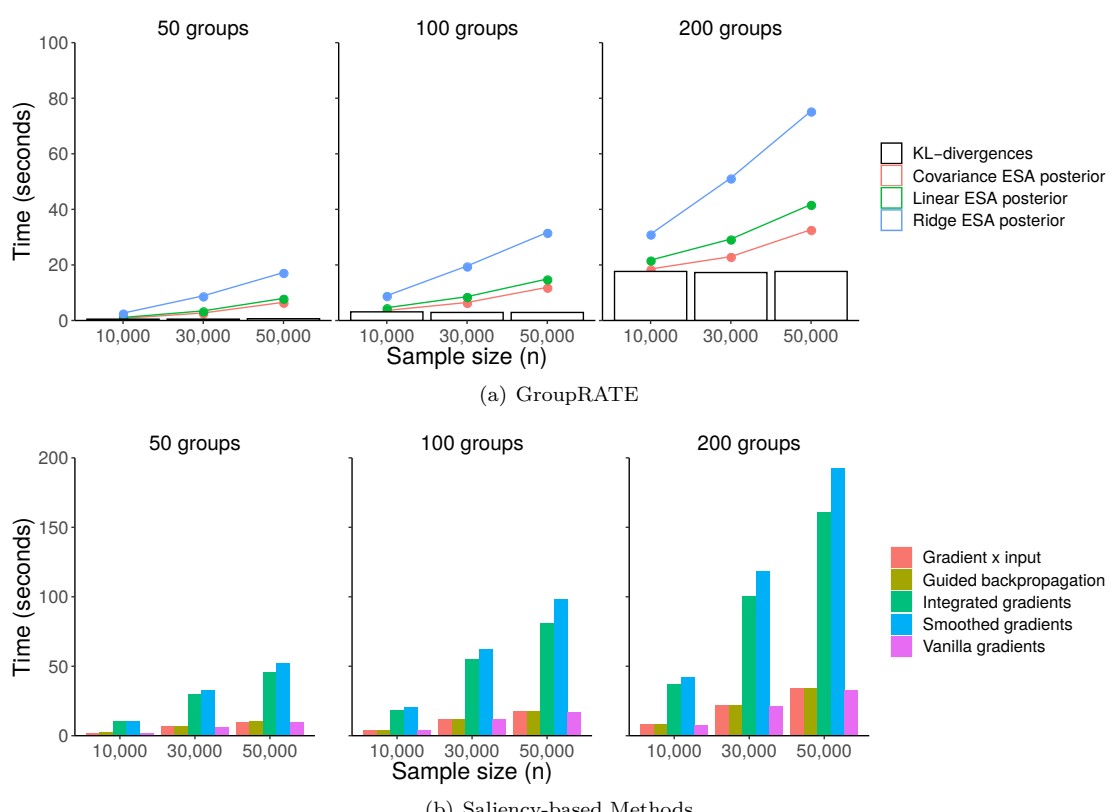

(a) GroupRATE

(b) Saliency-based Methods

Figure 5: Mean empirical computation times for the different group-level variable importance methods across 100 replicates. Panel (a) depicts the time it takes to run the GroupRATE framework while using different effect size analogue projections. The bars denote the time to compute the KL-divergence posterior, while the lines indicate the total GroupRATE calculation (effect size analog (ESA) posterior estimation plus the KL-divergence computation). Panel (b) shows the computational time that it takes to run competing approaches with parallelization while using 32 threads. The random forest mimic is excluded as its computation time is 1-2 orders of magnitude larger than the saliency methods ($G = 50$ with $N = 1 \times 10^4$ takes 460 seconds, and $G = 200$ with $N = 5 \times 10^4$ takes over 2.5 hours).

Overall, we found that a large number of the highly ranked genes identified by GroupRATE with the covariance projection have previously been identified by past publications as having some functional relationship with the traits of interest. For example, GroupRATE ranks the genes *Dnah8* and *Rsph1* on chromosome 17 as being the top two most enriched for the percentage of CD8+ cells in mice (Figure 6). This same region has been reported by multiple functional studies as having highly significant quantitative trait loci (Stefansson et al., 2007; Winkelmann et al., 2007) and has been identified by many computational methods have as having variants that contribute to non-additive variation for CD8+ cells (Crawford et al., 2019; Demetci et al., 2021). Valdar et al. (2006) also reported finding the most significant non-additive effects for immunological phenotypes (including percentage of CD8+ cells) around the major histocompatibility complex (MHC) on chromosome 17. Similarly, GroupRATE prioritizes *Rarb* on chromosome 14 as being the top ranked genes for body length. For context, *Rarb* has an orthologous gene in humans which contains common body mass index-associated variants that confer risk of extreme obesity (Cotsapas et al., 2009).

In these data, using GroupRATE with the different effect size analogues led to identifying different enriched genomic regions for both body length and the percentage of CD8+ cell traits. We believe that this is most likely due to the theoretical properties underlying their respective projection operators. First, performing group-level importance using the generalized inverse effect size analogue failed to yield any distinct gene rankings in either phenotype. We hypothesize that this is largely due to the Bayesian neural network and the least squares projection struggling to discern between associated features in the presence of high collinearity (see Appendix A). In contrast, regularization via the ridge is designed to select no more than a few variants in a given correlation block (Hoerl & Kennard, 1970). While this leads to better identification of signal than the generalized inverse (which has no penalization term), it still does not prioritize all trait-relevant genes. For example, in body length, GroupRATE with the ridge projection does not give any high importance to genes on chromosome 2 which has been shown to play a significant role in the genetics of growth, body weight, and body composition in mice (Yi et al., 2004; 2006; Lembertas et al., 1997; Jerez-Timaure et al., 2004; Vitarius et al., 2006; Ankra-Badu et al., 2009). The covariance operator, on the other hand, will compute analogue estimates based on the true effect size among all correlated variants in a gene boundary (again see Appendix A). This strategy is also not perfect in all cases. To see this, in the body length trait, performing GroupRATE with the covariance operator failed to highly prioritize any genes on the X chromosome — which is interesting because the X chromosome is well known to strongly influence adiposity and metabolism in mice (Chen et al., 2012). Nonetheless, in general, GroupRATE with the covariance operator was able to identify more significant genes associated with body length and CD8+ cell percentage.

### 5.3 Structural Brain Region Enrichment Analysis in MRI Scans

To demonstrate the application of GroupRATE on other types of data, we evaluated it on structural brain MRI. The images are available as part of the UK Biobank, which conducted a comprehensive study of 500,000 people recruited from the UK's general population between 2006 and 2010 (Miller et al., 2016). The participants aged between 40-69 years old and provided blood samples for biochemical tests, imaging, genotyping, as well as a wide range of self-reported information and physical measurements. The protocols for obtaining the different measurements from the participants have been published in the literature (Alfaro-Almagro et al., 2018).

Studies have shown that the difference between true brain age and the brain age predicted by a model can act as a biomarker for risk stratification and clinical applications (Cole et al., 2017; Hajek et al., 2019; Kolbeinsson et al., 2020). However, the best performing models in the literature are often neural networks which are difficult to interpret. Here, we apply GroupRATE to the analysis of brain age difference in the UK Biobank population to identify structural brain regions that highly associate with this biomarker.

This study utilized 12,022 3D images from a release accessible to researchers. The images employed were T1-weighted, which accentuate the difference between white and grey matter. A subset of 522 images were used for a held-out test set. At full $1\,\text{mm}^3$ resolution, the $182 \times 218 \times 182$ volumes are too large for efficient computation, therefore we applied GroupRATE to downsampled volumes. The downsampling operation was a learned convolutional layer with both kernel size and stride set to 7. This resulted in a smaller volume of $26 \times 31 \times 26$ to which GroupRATE can be efficiently applied. The 140 groups are defined from the brain region atlas provided as part of the UK Biobank imaging release. All images had been aligned to the MNI152

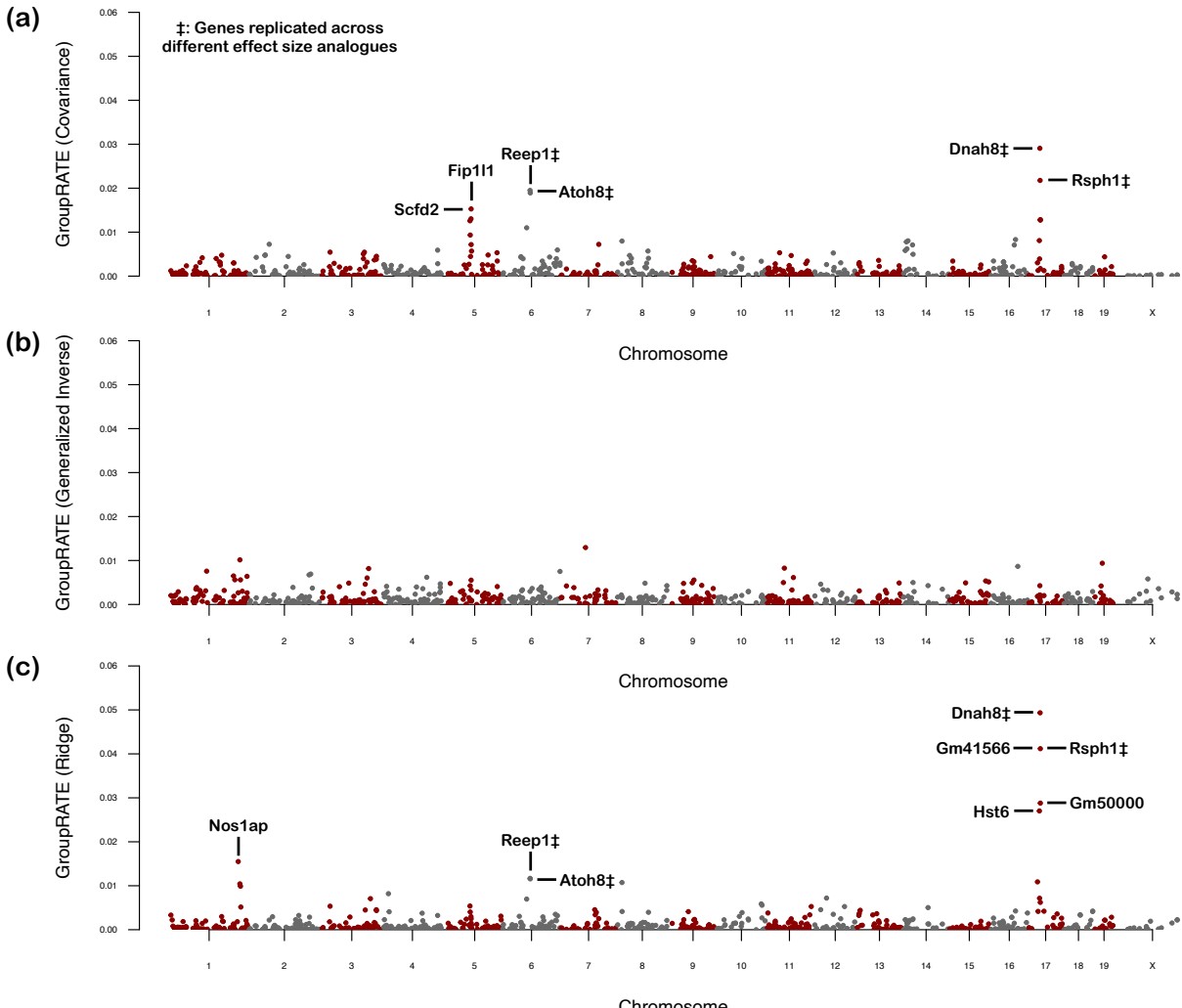

Figure 6: Group-level genome-wide scan for percentage of CD8+ cells in the heterogeneous stock of mice dataset. Here, GroupRATE variable importance is computed while using effect size analogues derived from the (a) covariance, (b) generalized inverse, and (c) ridge penalized projections. Chromosomes are shown in alternating colors for clarity, with the top notable genes being annotated on the plot near their genomic position. ‡: Genes that replicated as being highly prioritized while using different effect size analogues.

template, allowing for direct voxel-to-voxel mapping between the atlas and images. The $182 \times 218 \times 182$ atlas was downsampled using max-pooling (kernel size and stride both set to 7) to give a volume map with the same dimension as the MRI.

We fit a prediction model using a convolutional neural network with 3D ResNet-like blocks. The details of the non-Bayesian model architecture and training have been previously described in Kolbeinsson et al. (2021). The changes we made here were to replace the final layer with a Bayesian equivalent whose weights make use of variational inference. The training objective was equation 12 with $\eta = 10^{-4}$ optimized using Adam. The Bayesian layers were created using the `Bayesianize` library (Ritter et al., 2021).

The main results of the GroupRATE analysis are shown in Figure 7, while the complete results for all groups are shown in Figure S3. GroupRATE identifies a number of brain regions that have associations with increased brain age difference as shown in previous studies. The highest ranked region was the left planum temporale, which is thought to be neurologically connected to language (Binder et al., 1996; Wernicke, 1874).

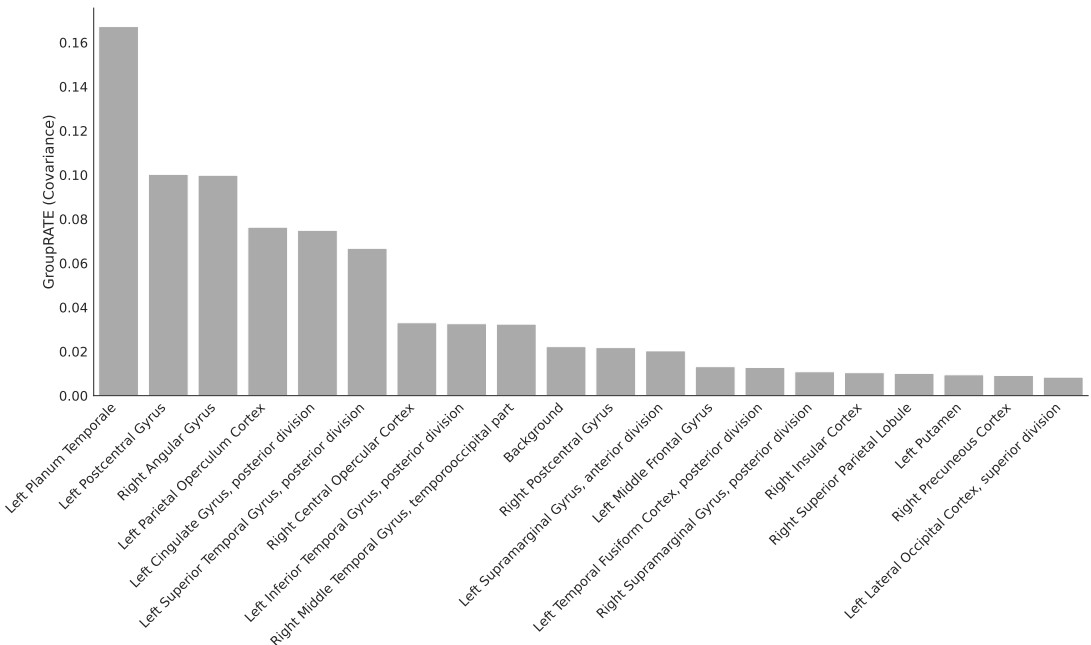

Figure 7: Ranked importance of the top 20 groups identified by GroupRATE using the covariance effect size analogue. A subset of the 140 total regions have notably higher importance. Most of these groups have been described previously as having associations with brain development, ageing or disorders.

Furthermore, changes to its volume have been associated with schizophrenia (Kwon et al., 1999). Gyri regions were particularly highlighted with the left postcentral gyrus and the right angular gyrus being the second and third highest ranked region, respectively. Previous studies on brain structure changes with age have also found both of these regions to reduce in volume with increased age (Sussman et al., 2016), demonstrating that GroupRATE is selecting structurally-relevant regions.

## 6 Discussion

In this paper, we developed a novel group-level global interpretability method for Bayesian neural networks. Here, we focused on settings in which collections of predictor variables are intrinsically meaningful and the goal is to rank these groups of features based on their scientific relevance. We worked in a very flexible variational Bayes approach to deep learning and proposed a sample covariance operator to develop an effect size analogue for the input variables of a neural network. Next, we extended the recently proposed RelATive cEntrality (RATE) measure (Crawford et al., 2019) to our setting, provided closed-form solutions for its implementation, and developed the GroupRATE criterion. Lastly, we illustrated the performance of our framework in a thorough simulation study and in broad real data applications including statistical genetics and biomedical imaging. Our method outperforms or achieves performance on par with the state-of-the-art, while avoiding the need for a separate and (often) time consuming tuning step.

In its current form, we have focused on demonstrating the utility of GroupRATE with a particular Bayesian neural network where only the weights on the outer layer are considered as random variables (see again Figure 1). Note, however, that we are not restricted to this architecture and each of the innovations we have presented can be applied to any deep learning method that provides a notion of uncertainty over the predictions. The effect size analogue is merely a multivariate summary statistic which can be derived after fitting any model. This means that, as long as one has access to empirical estimates of its posterior distribution, relative centrality measures can always be computed. While the variational Bayes framework described in our work gives an exact Gaussian posterior over $f$, many recent works have focused on calculating approximations to the posterior of an already-trained deterministic network using Laplace approximations (Ritter et al., 2018) or

stochastic gradient descent iterates (Maddox et al., 2019). Combining these approaches with the GroupRATE framework would allow variable importance calculations to be performed on an already trained deterministic network (without the need for retraining with a mean-field variational posterior on the final layer).

Much of this study was motivated by the increasing popularity of nonparametric predictive modeling (particularly with neural networks) in biomedical applications. As long as such methods continue to be applied in areas where interpretability is a requirement, *post-hoc* methods such as Group(RATE) will have utility. Rudin (2019) suggest that a better modeling approach is to place interpretability at the heart of model building from the beginning of a project. Fortunately, Bayesian neural networks offer this possibility through the use of sparsity inducing priors (van Bergen et al., 2020; Song & Li, 2021; Chen et al., 2020; Kassani et al., 2022; Lu et al., 2018; Feng & Simon, 2017; Fortuin, 2022; Ghosh et al., 2019; Cheng et al., 2022) or from constructing partially connected network architectures that are based on biological annotations or scientific knowledge (Demetci et al., 2021; Elmarakeby et al., 2021; Bourgeais et al., 2021). However, this approach is extremely challenging for problems with very little *a priori* knowledge and so *post-hoc* interpretation methods are likely to remain useful in practice for the foreseeable future.

### Author Contributions

JIH, DU, SF, LC, and SF conceived the study and developed the methods. JIH, DU, and LC developed the algorithms and software. JIH, DU, AK, and KS performed the analyses. All authors wrote and revised the manuscript.

### Software Availability

Software for implementing the Bayesian neural network framework with RATE and GroupRATE significance measures is carried out in `R` and `Python` code, which is available at `https://github.com/lorinanthony/RATE`.

### Acknowledgments

This research was supported in part by grants P20GM109035 (COBRE Center for Computational Biology of Human Disease; PI Rand) and P20GM103645 (COBRE Center for Central Nervous; PI Sanes) from the NIH NIGMS, 2U10CA180794-06 from the NIH NCI and the Dana Farber Cancer Institute (PIs Gray and Gatsonis), an Alfred P. Sloan Research Fellowship, and a David & Lucile Packard Fellowship for Science and Engineering awarded to L. Crawford. S. Filippi was partially supported by the EPSRC (grant EP/R013519/1) and J. Ish-Horowicz gratefully acknowledges funding from the Wellcome Trust (PhD studentship 215359/Z/19/Z). D. Udwin was a trainee supported under the Brown University Predoctoral Training Program in Biological Data Science (NIH T32 GM128596). Lastly, this research was conducted using the UK Biobank Resource under Application Numbers 14649. Any opinions, findings, and conclusions or recommendations expressed in this material are those of the author(s) and do not necessarily reflect the views of any of the funders.

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
