# OpenReview forum: "A Group Variable Importance Framework for Bayesian Neural Networks"
_TMLR — Rejected by TMLR_

### Review · Reviewer_HwXW · 2023-03-14

**Summary Of Contributions:**

This work begins with an existing variable importance measure, RATE, and extends it in two ways. The first extension is to allow it to operate on Bayesian neural networks. The second is to allow it to operate on variable groups (rather than individual variables). The authors also extend approximations for RATE to this setting, and demonstrate the use of their method on two real and several synthetic datasets.

**Audience:**

Yes

**Broader Impact Concerns:**

This work presents no ethical issues.

**Claims And Evidence:**

No

**Requested Changes:**

Due to the experiments lacking sufficient baselines and leaning towards qualitative evaluations, and the proposed method not offering clear performance advantages relative to a simple baseline, I'm marking the Claims & Evidence question as "no" at this time. But I would be willing to change that rating if the authors improved their experiments.

The discussion above lists many points that could be improved, including clarifications to the method, related work, and specific requests for improvements in the experiments.

**Strengths And Weaknesses:**

## Strengths

Understanding important variables in complex systems, particularly when they are modeled with models like neural networks, is an important problem. Local interpretation (explaining individual predictions) has received more focus than global interpretation as of late, so this is a worthwhile problem to focus on. And this work considered an approach that, to my knowledge, has not been explored to a great extent in the literature. The proposed approach initially seems like it could be very computationally costly, but the authors present several reasonable approximations to make it tractable.

## Weaknesses

I noticed a number of ways in which the paper had weaknesses that could be improved, so I'll enumerate these below.

Method:
- The premise of this method is that we should be interested in the posterior distribution over coefficients $\beta$, and that a feature is important if its coefficient is strongly related to the coefficients of the remaining features. As the authors show, this can be calculated via a KL divergence measure, or the average of this measure across coefficient values (which is equivalent to the mutual information). In my view, it's not obvious why this is a useful measure of importance to the predictive task. An alternative notion of importance, which is perhaps more natural, is whether knowing the variable's value is important to make an accurate prediction. I wonder if the authors could expand on why their basic approach makes sense, because it's a fundamental point for the paper.
- In discussing the distinction between the KL divergence measure and mutual information, the authors point out that a variable can achieve a high mutual information score even if its mean coefficient is very small or near zero. They then say that their KL divergence measure corrects this possible issue: "this is in contrast to the RATE criterion which also takes the posterior mean (or marginal effect) of input features $\mu$ into account when determining variable importance." I agree that the KL divergence measure will not allow features with zero marginal effect ($\mu_j = 0$) to have high importance. But distinguishing between 1) the marginal effect and 2) the posterior relationship with other coefficients raises an interesting question, which is whether 2) is actually important. Note that under the KL divergence measure, a feature will be deemed unimportant if it is unrelated to the remaining coefficients, *even if it is marginally very important* - but is this desirable? I wonder if the authors could expand on this, and if they could add as an ablation/baseline a method that simply calculates $\mu_j^2$ as a score for each variable (with a corresponding generalization to groups, perhaps $||\mu_g||^2$ or $||\mu_g||^2 / |A_g|$).
- The authors motivate this extension of RATE as allowing it to operate on Bayesian neural networks (BNNs). But it appears that the technique only works on a specific type of BNN, where all layers are deterministic until the final layer, in which only the weight matrix is probabilistic (and the bias remains deterministic). I see why this formulation is necessary, because it simplifies the distribution over predictions and enables a simple Gaussian distribution over the covariance estimates, but it's clearly somewhat restrictive. BNNs are not usually trained in this fashion, so this method appears to require using an essentially custom architecture. It might be helpful for the authors to expand on the generality of their approach, because the paper perhaps incorrectly suggests that the method is highly general.

Related work:
- In discussing existing approaches for understanding neural networks, the authors describe one approach that amounts to fitting a random forest (or GBM) model as a surrogate, and then using a variable importance measure designed for that model. As one example, they describe the permutation importance measure introduced by Breiman (2001). This method is equally applicable to neural networks, why would it be necessary to train the surrogate tree-based model? In fact, this method probably should have even been included as a baseline.
- Related to the permutation importance measure: there's a wide class of variable importance methods that are overlooked here, beyond those based on input gradients. Permutation importance is just one method where features have their information removed in some sense, and we measure the resulting impact on a model's output or loss. Here are two papers that review such methods when applied to global importance [1] and when applied to both local and global importance [2]. Note that these measures can involve retraining models with different subsets of features, or retraining the model with different subsets [3] or with single features held out [4].
- The authors also neglect to mention existing work that sparsifies neural networks to identify important input features. This can be done using group sparse penalties and training with proximal methods, see [5, 6, 7], or by training with differentiable gating mechanisms [8, 9]. The authors wrote "to our knowledge, grouped variable importance has not yet been studied for neural networks, despite several analogous works for other supervised models" - but the aforementioned approaches are easy to extend to operate on pre-defined feature groups (e.g., [10] did so for time series models).
- Related to a comment above: the authors describe their technique as being applicable to BNNs, but it seems to be applicable only to a specific and unusual type of BNN - one with probabilistic weights in just the final layer. If the authors believe this to be a common BNN approach, they might mention prior work that uses this formulation.

[1] Covert et al, "Understanding global feature contributions with additive importance measures" (2020)

[2] Covert et al, "Explaining by removing: A unified framework for model explanation" (2021)

[3] Williamson & Feng, "Efficient nonparametric statistical inference on population feature importance using Shapley values" (2020)

[4] Lei et al, "Distribution-free predictive inference for regression" (2018)

[5] Feng & Simon, "Sparse-input neural networks for high-dimensional nonparametric regression and classification" (2017)

[6] Dinh & Ho, "Consistent feature selection for analytic deep neural networks" (2020)

[7] Lemhadri et al, "Lassonet: Neural networks with feature sparsity" (2021)

[8] Abid et al, "Concrete autoencoders: Differentiable feature selection and reconstruction" (2019)

[9] Yamada et al, "Feature selection using stochastic gates" (2020)

[10] Tank et al, "Neural Granger causality" (2021)

Experiments:
- The simulation setting in section 5.1 is unusually complex, I wonder if it would be worth simplifying it slightly. We need the variation in number of groups, group size, number of samples, etc. But I'm not sure it's important for $\Sigma_{bg}$ to be sampled rather than fixed, for $\Sigma_{grp}^g$ to be sampled rather than fixed, or for the model to explain a fixed portion of the variance. I also would have been interested in seeing results with more extreme contributions from the non-additive interaction terms (if I understand correctly, $\rho = 0.1$ and $\rho =0.9$).
- The reported values of $\rho$ in the bullet points of page 10 seem to contradict those that are listed immediately after when describing Scenario I and Scenario II (0.4 and 0.6 vs. 0.75 and 0.5).
- The network architecture, with just 8 hidden units, seems strangely small for a dataset with $J = 1000$ variables. Why did the authors make this choice, and how would the results change if they used an architecture with, for example, two hidden layers of 32 or 128 units? It would be unfortunate if the GroupRATE method didn't work as well when the number of hidden units in the final layer was large, but this is important to verify.
- As mentioned above, there are several baseline methods that the authors could have added. These include permutation importance, SAGE, and perhaps mean absolute SHAP values (another common heuristic for generating global importance scores from local importance scores). Note that with removal-based methods like permutation tests, it is trivial to run the method directly using feature groups, so a heuristic group aggregation step is not necessary.
- The results on the synthetic experiments seem to indicate a minimal difference between the proposed method(s) and the simplest baseline, vanilla gradients. Vanilla gradients may even perform better in several cases. This seems to undermine the theory that RATE/GroupRATE provides a uniquely valuable variable importance measure.
- The real data experiments are, understandably, somewhat qualitative in their evaluation. I wonder if the authors could provide any quantitative performance measures for the real data experiments, because the only quantitative comparison currently provided is for the synthetic datasets. For example, would it be possible to train models with different numbers of highly ranked feature groups and compare the predictive accuracy?
- The experiment involving structural MRIs is interesting methodologically, but the verification of results is basically missing. Is there anything more definitive the authors can provide here?

---

> ### Author Response · Authors · 2023-05-02
> **Answer to Reviewer HwXW -- part 1**
>
> We would like to thank the reviewer for the detailed comments and will address their questions/comments in the following paragraphs.
>
> ### Justification of the variable importance method
>
> Point #1: We thank the reviewer for bringing up this important point. Indeed, another natural measure for variable importance is investigating whether a variable’s value is important to make an accurate prediction. This approach is actually a focus of previous works in the literature (e.g., Paananen et al. AISTATS 2019). There are a few important distinctions between these works and the ones we present here. First, most approaches that assess the sensitivity of predictive performance with respect to a set of features do so with the ultimate goal of performing locally interpretability where the aim is to explain specific classification/predictions decisions within a given sample. Here, our goal is to assess global interpretability where we want to rank/select features based on their contributions to overall variation in an observed population. Second, our work is motivated by applications where there are multicollinear features but only a few are truly important. The disadvantage of the predictive sensitivity approach is that multiple correlated features could lead to the same loss in performance even if they are not all truly associated with the outcome of interest. The advantage of our approach is having the ability to account for those complex feature relationships. In the case where we have two highly correlated variables, our KL approach will only identify both variables as being important if they are each associated with the output of interest.
>
>
> Overall, we believe that our basic approach makes sense because one can think of sets of correlated variables as being part of a network (e.g., groups of genes in a regulatory system or patches of pixels in the region of image). Typical questions in network studies simplify to the general issue of determining the “centrality” of nodes—the potential importance of individual components in relation to the other nodes in the entire network. In the nonlinear regression context values an effect size analogue β close to zero may be interpreted as “null hypotheses” with little to no relevance to the modeled outcome. Therefore, searching for the most central (i.e., influential or important) features simply reduces to looking for the greatest KL when setting each β = 0. In conventional statistics, our proposed variable selection procedure is very much related to precision analysis. It follows that the rate of change for the KL (i.e., the first derivative of equation (6) with respect to a given effect size analog) is found via the term δ_j. This means that the closed form computation of the KL is directly impacted by the deviations between the approximation of a given predictor’s posterior mean and the assumption that its true effect is zero. Therefore, δ_j characterizes the implied linear rate of change of information when the effect of the j-th feature is absent from the model—thus, providing a natural (nonnegative) numerical summary of the role of the j-th coefficient β_j in the multivariate posterior distribution.
>
>
> Point #2: We also thank the reviewer for bringing up this interesting question. Indeed, the utility of selecting features that are both (1) marginally important and (2) share a posterior relationship with the other coefficients is likely to be application specific. In the context of both genomics and medical imaging, we believe that it is desirable to select features that are both marginally important and also contain information about the effect of other features. In genetics, it is unlikely that an important gene will act in isolation without having a covarying effect on other genes with similar mechanisms. A similar idea can be stated for images where we want to identify subsets of features that characterize regions of interest. We appreciate the idea of adding an ablation/baseline that simply computes μ_j (or ||μ_j||^2 for groups). In the simulation study of our revision, we will add these to highlight the utility of our specific variable importance KL strategy.

---

> > ### Author Response · Authors · 2023-05-02
> > **Answer to Reviewer HwXW -- part 2**
> >
> >
> > ### Structure of the Bayesian Neural Network
> >
> >
> > In general, GroupRATE can be applied to any Bayesian neural network, however if the effect size analogue posterior is non-Gaussian then the KL-divergences are computed using sampling. We did not cover this in the text as we felt there was sufficient novelty in the work presented using the proposed last layer Bayesian architecture, but we would be happy to add a section describing how GroupRATE could be applied using sampling strategies in the cases where the effect size analogue posterior is non-Gaussian.
> >
> >
> > While the specific last layer Bayesian architectures we discuss in the main text use a deterministic bias for the final layer, GroupRATE can also  be computed in closed-form if they are Gaussian-distributed. Some recent work has investigated using these “last layer Bayesian” networks in an effort to decouple the feature representation learning performed by the inner layers and the uncertainty quantification of the final prediction and has shown to lead improved uncertainty calibration and out-of-distribution detection in some settings [1,2,3,4]. We will add these examples of possible applications to the text.
> >
> > ### Related work, experiments and comparison
> >
> > Thank you for bringing these relevant works to our attention. We feel that the most directly comparable are SAGE and grouped permutation importance and so will update our experiments to include these comparisons, with the remaining methods cited throughout the main text. We want to make sure we note that, given the scope and the mission of TMLR as a journal, we do not believe that the final decision of our paper should be affected by the competitiveness of GroupRATE relative to these methods. Rather, we want these additional experiments to show a more comprehensive view of the advantages/disadvantages between our proposed approach and those that exist in the literature. To that end, in the revised manuscript, we will also update the experiments to include larger network architectures, which we have used with GroupRATE in the past despite restricting ourselves to very simple architectures in the main text. The architecture used in the current version of the paper was selected as the simplest one that led to reasonable test set performance — given that our main focus and motivation for this work is the interpretation of a trained model rather than selecting an optimal trained model.
> >
> >
> > We will also add a comparison to the effect size analogue posterior mean, although this approach will likely suffer from similar drawbacks as any variable-level scoring method in that the optimal choice of aggregation function (e.g. median, max, mean, …) is not trivial.
> >
> > ### References
> > [1] Carlos Riquelme, George Tucker, and Jasper Snoek. Deep Bayesian bandits showdown: An empirical comparison of Bayesian deep networks for Thompson sampling. ICLR, 2018.
> > [2] Watson, Joe, et al. "Latent derivative Bayesian last layer networks." International Conference on Artificial Intelligence and Statistics. PMLR, 2021.
> > [3] Brosse, Nicolas, et al. "On last-layer algorithms for classification: Decoupling representation from uncertainty estimation." arXiv preprint arXiv:2001.08049 (2020).
> > [4] Kristiadi, Agustinus, Matthias Hein, and Philipp Hennig. "Being Bayesian, even just a bit, fixes overconfidence in ReLU networks." International conference on machine learning. PMLR, 2020.

---

> > > ### Comment · Reviewer_HwXW · 2023-05-04
> > > **Response**
> > >
> > > Thanks to the authors for their response. I can comment on a couple points made in the rebuttal.
> > >
> > > ### Justification of the method
> > >
> > > > First, most approaches that assess the sensitivity of predictive performance with respect to a set of features do so with the ultimate goal of performing locally interpretability where the aim is to explain specific classification/predictions decisions within a given sample. Here, our goal is to assess global interpretability where we want to rank/select features based on their contributions to overall variation in an observed population.
> > >
> > > That's not exactly true – one of the oldest and simplest model interpretation methods, Breiman's permutation test, analyzes the sensitivity of predictive performance for the purpose of global interpretability. I'm not sure the choice of analyzing the posterior dependence of coefficients is justified by the need for global interpretation, because the predictive sensitivity is equally applicable to the global case.
> > >
> > > > Second, our work is motivated by applications where there are multicollinear features but only a few are truly important. The disadvantage of the predictive sensitivity approach is that multiple correlated features could lead to the same loss in performance even if they are not all truly associated with the outcome of interest.
> > >
> > > This is a legitimate drawback of the predictive sensitivity approach, but it may not be a fatal flaw: identifying multiple correlated variables as relevant to the outcome is in some sense correct if they all provide knowledge (in an information-theoretic sense) about the response.
> > >
> > > On the other hand, consider a failure mode of the proposed approach: if the posterior distribution does not involve strong dependencies between variables (e.g., they are independent in the posterior for $\beta_{cov}$), this method would assign them all zero importance regardless of the association strength. It seems like this could occur in a dataset with independent features, or even one independent feature that's strongly associated with the response. That may be a more significant flaw, so the new experiments the authors discussed will be helpful to test whether the association strength alone is a useful variable importance indicator.
> > >
> > > > Overall, we believe that our basic approach makes sense because one can think of sets of correlated variables as being part of a network (e.g., groups of genes in a regulatory system or patches of pixels in the region of image).
> > >
> > > But the goal here is to assess importance for a specific response, not interdependence with other variables or centrality in a network. Consider the following: there could be a set of variables with only a weak association with the outcome, but which are strongly related according to the posterior distribution. These could then be deemed very important by the proposed measure, which might be misleading. Wouldn't this behavior be possible to induce with a particular data distribution?
> > >
> > > ### Bayesian neural network assumptions
> > >
> > > Both of the updates the authors proposed would be helpful: 1) citing works that use this last-layer Bayesian approach (and discussing its advantages, which perhaps go beyond reducing the number of parameters), and 2) adding a section describing how the proposed approach works with a fully Bayesian neural network.
> > >
> > > ### New experiments
> > >
> > > The proposed experiments sound like they would be very helpful for this work.
> > >
> > > > We will also add a comparison to the effect size analogue posterior mean, although this approach will likely suffer from similar drawbacks as any variable-level scoring method in that the optimal choice of aggregation function (e.g. median, max, mean, …) is not trivial.
> > >
> > > I'm not sure I understand the issue. Wouldn't this just involve analyzing $\mu$, why is there any need for an aggregation function? If the authors mean how to handle the feature grouping, their proposal of normalizing the importance by the group size seems like a sensible choice, so perhaps $||\mu_j|| / |A_j|$.

---

> > > > ### Author Response · Authors · 2023-05-06
> > > > **Answer on the method and the comparison to the effect size analogue posterior mean**
> > > >
> > > > We want to thank the reviewer for engaging in dialogue with us on our manuscript. The point about failure modes for the RATE and groupRATE framework is a really good and important one. In fact, we would argue that there are two extreme cases to consider with respect to RATE and groupRATE. The first is as the reviewer mentioned: in the setting where the covariance of the effect sizes for two variables is truly (and exactly) the identity, the \delta term of the RATE metric will go to zero regardless of the magnitude of the effect. The other extreme situation worth considering is in the limiting case where two variables are perfectly correlated with the response. Since they each explain exactly the same amount of information, the RATE metric for each variable will also be identical.
> > > >
> > > > In the revised manuscript, we will be sure to mention these situations. We also want to acknowledge that both of these extreme cases rarely would happen in practice; the covariance between effects will almost never be exactly the identity, and variables could be highly correlated but are less likely to be perfectly correlated. For example, the case of independent variables is considered in the original RATE paper by Crawford et al. (2019) (Section 4.1.1, Proof of concept simulations) which shows that RATE can capture associations under this scenario. Nevertheless, we agree with the reviewer that it will strengthen the understanding of our approach if we explicitly address these issues in the paper.
> > > >
> > > > Regarding the comparison to the effect size analogue posterior mean, we indeed meant aggregation in terms of handling the feature groupings. Multiple approaches could be used there. Following your suggestion, we have re-run the experiments using $||\mu_j|| / |A_j|$ as a group importance measure and the results show that GroupRATE values have higher AUCs. Accordingly, we will add these results to the revised manuscript as they show the added value of considering the covariance structure of the ESA posterior rather than just the marginal effects

---

### Review · Reviewer_3a83 · 2023-04-01

**Summary Of Contributions:**

In this paper, the authors consider the challenge of interpreting neural networks. While traditional variable importance methods provide local explanations for specific predictive decisions, they often generate false positives and negatives in high-dimensional and collinear data settings. The authors aim for global interpretability by identifying essential groups of variables through aggregating univariate signals, ultimately enhancing power and reducing false discoveries. So, the authors propose a method called GroupRATE, which extends the recently proposed RATE measure to the Bayesian deep learning setting, leveraging partial covariance structures and variable interactions to assess the group-level significance of features. GroupRATE employs an information-theoretic metric on the joint posterior distribution of effect sizes to evaluate the group-level significance of features, without needing tuning parameters, which can be challenging and expensive to select. The authors demonstrate the effectiveness of their framework using both simulated and real data.

**Audience:**

Yes

**Claims And Evidence:**

No

**Requested Changes:**

1. In what ways could the proposed GroupRATE algorithm be further differentiated from the existing RATE method to demonstrate its innovation?

2. Could the authors provide additional justification for the choice of the simulation environment and its relevance to real-world scenarios?

3. Are there plans to expand the experiments to include other types of data, such as image inputs, to showcase the method's broader applicability in various domains?

**Strengths And Weaknesses:**

Pros:

1. The paper introduces a novel algorithm for identifying important groups of variables in neural networks, contributing to the field of global interpretability.

2. The GroupRATE criterion addresses the limitations of traditional variable importance approaches by focusing on group-level significance of features, rather than individual ones.

3. The method does not require tuning parameters, which can be time-consuming and difficult to select, simplifying the process for users.

4. The authors demonstrate the utility of their framework on both simulated and real data, showcasing its potential applicability in various scenarios.


Cons:

1. The paper's readability could be improved, as the technical novelty and comparisons with existing algorithms are not easily understood.

2. The proposed algorithm is an extension of the existing RATE method, which might make it seem less innovative.

3. The simulation environment appears artificial, and it would be beneficial if the authors could justify its relevance to real-world situations.

4. Although the authors claim that their approach can be applied to various neural network architectures, the results are limited to a simple neural network, warranting further exploration and validation.

5. The experiments focus exclusively on tabular data sets, and it would be advantageous to include cases with image inputs to demonstrate broader applicability.

---

> ### Author Response · Authors · 2023-04-28
> **Answer to reviewer 3a83**
>
> We would highlight two substantial contributions that go well beyond the existing RATE method, while maintaining its novelty as a global explanation method focused on interactions among variables. The two contributions are: (1) efficient application to Bayesian neural networks and (2) the formulation of an elegant solution to the “group” explainability problem. While it is true that these extensions are to some extent straightforward, taken together they fill what we see as a substantial gap in the literature: explainable ML is a very large area, but the focus on prespecified groups of variables is only recently emerging. Similarly, there are many more local explanation methods than global methods, and few methods specifically focus on the case of Bayesian neural networks. Taken as a whole, we see the GroupRATE algorithm as a very practically useful approach for the applied scientist.
>
> The simulation environment was chosen to mirror real world data structures that occur in both genomics  and medical imaging applications. In the former case, it is well known that genetic variants that are in close proximity along the genome have stronger correlation than those that are further away. This concept is called linkage disequilibrium or LD, and effectively creates a block-like variance-covariance structure between features in the data (e.g., much like what is depicted in the current Figure 2). Oftentimes, these LD blocks also correspond to natural feature groupings along the genome (e.g., genetic variants being annotated for genes or other regulatory units). Controlling the balance between the contribution of additive and non-additive effects is also meant to mirror phenomena that occur in real genomic data sets. Many association mapping studies in humans have shown that gene-by-gene interactions can notably contribute to trait variation (e.g., Sheppard et al. PNAS 2021). Nonlinear Interactions are also well-known contributors to trait architecture in several model organisms including mice and drosophila (e.g., Spierer et al. PLOS Genetics 2021). The v^2 parameter in our simulation environment determines how much variance in the generated response is due to signal versus noise (this is the concept of heritability in genetics), while ρ is a mixture parameter which determines how much of the signal is driven by linear versus nonlinear effects. We choose values for v^2 and ρ to mirror realistic values seen in the literature.
>
> Our final real data example is from medical imaging (brain MRIs), which is an important application of deep learning in the biomedical literature and one in which groups of variables (voxels) have a more meaningful interpretation than individual variables. Furthermore, similar to the genomics application, pixels (or voxels) that are spatially closer together in images are likely to have tighter correlation structures than those that are relatively further away. This also helps to motivate the block variance-covariance structure we use to simulate data. Overall, we therefore feel that this example is well-suited for analysis using GroupRATE, however we appreciate that the difficulties associated with designing simulation studies for image data mean that our investigations into image data are less extensive than for genomic data.

---

### Review · Reviewer_JNGm · 2023-04-20

**Summary Of Contributions:**

Overall, the paper is very well and clearly written. This paper is dealing with an important question in deep learning, which is the interoperability of neural networks at the global level. Motivated by the natural group structure of SNPs data, this paper is focusing on identifying groups of variable importance in order to improve power and reduce the false discovery rate.  To deal with the challenge, the author has extended “RelATive cEntrality” (RATE) measure to the Bayesian deep learning under a group setting to allow for group feature ranking. The main contribution is of the paper is to identify groups of important predictor variables given a trained neural network.

In Section 3, the authors provide fundamental background knowledge about Univariate Variable Importance using Relative Centrality Measures so that it is natural to extend this to a group setting in Section 4. In Section 4, the authors also provide variational inference derivations of how this group extension is possible with the clear result in Equation 18 to describe the group variable importance.

This is a great paper and I recommend acceptance of the paper.

**Audience:**

Yes

**Broader Impact Concerns:**

No Broder Impact Concern.

**Claims And Evidence:**

Yes

**Requested Changes:**

Here I suggestion the following potential changes:

1. In the related literature, current interpretability for deep learning attracts attention from different perspectives. For example, Lu, Yang, et al. "DeepPINK: reproducible feature selection in deep neural networks." Advances in neural information processing systems 31 (2018) and "Deep-gknock: Nonlinear group-feature selection with deep neural networks." Neural Networks 135 (2021): 139-147 which deal with global univariate feature selection and group-level feature selection with controlled false discovery rate is worth discussing in the related work since these knockoff-based work has guaranteed false discovery control and the work is also motivated for addressing gene expression data challenges.

2. In terms of the experiment, I will suggest a setting with a comparison with competing methods to see direct power and fdr performance using the proposed method.

3. Description or an experiment to show the method's performance when rho allows for the bigger nonlinear component.

4, Will a strong correlation within each group of the variables affect much on the performance of the method?




**Strengths And Weaknesses:**

Strengths:

Nearly all aspects of the paper are well explained with intuition instead of just providing mathematical derivations. From Section 3 to Section 4, the mathematical derivation for individual variable importance in gamma_j before Equation 4 provides fundamental understanding when we extend this to a group setting in Section 4.4.

Moreover, the computational complexity for competing methods and the proposed method have been analyzed which shed light on the empirical results. The method is also demonstrated using two real-world high-dimensional dataset in Section 5.2 and Section 5.3. Software packages are also provided for reproducibility concerns and wide applicability of the paper for other readers to use.

Weakness:

In the Simulation studies of Section 5, the authors only compare the proposed method with competing methods using AUC. However, in the motivation of the paper, the authors mention that they target has high power and a low false discovery rate. If it is not a big computational burden, I will suggest the authors to compare both the power and false discovery rate (FDR) for their methods and the competing methods. I do not suggest to try all three different sample sizes, but maybe just using one sample size setting to showcase the power and fdr.  In the group-setting, a group-FDR has been proposed by paper "Deep-gknock: Nonlinear group-feature selection with deep neural networks." Neural Networks 135 (2021): 139-147.

For the contribution of additive effect parameter p, the authors uses rho= 0.4 and 0.6 close to 0.5. I am wondering what happens if rho takes a value of 0.9 with a more extreme nonlinear case, will the proposed method also dominates other competing method.

---

> ### Author Response · Authors · 2023-04-28
> **Answer to reviewer JNGm**
>
> We appreciate your positive review of our work. Thank you for highlighting DeepPINK and Deep-gknock; we will cite and discuss them in the revised version of the manuscript along with other papers suggested by the other reviewers. We are also happy to investigate the performance of the approach in the presence of an increasing non-linear component in the response as well as report the power and FDR of the proposed method and the competing approaches for a given threshold.

---

### Comment · Action_Editors · 2023-04-20
**Rolling discussion**

Dear authors,

Reviews are submitted. I am writing to let you know that you have two weeks to do rebuttals and rolling discussions before the reviewers submit their final recommendation.

Best wishes,
Tongliang

---

### Decision · Action_Editors · 2023-06-04

**Recommendation:** Reject

**Comment:**

Considering the concerns brought up during the review process that demand substantial revisions, I find myself leaning towards a decision of "Reject". This decision, however, should not discourage the authors as it is primarily driven by the lack of a "Major Revision" option in the system. I would, strongly urge the authors to address two major issues and consider resubmission.

The first issue pertains to the claimed broad applicability of the proposed method. The empirical evidence provided by the authors, which has been tested on specific network architectures and datasets related to genomics and medical imaging applications, does not convincingly validate these claims. As such, questions regarding the generalizability of this method across a variety of network architectures and datasets remain unanswered.

The second concern is a theoretical one raised by Reviewer HwXW. They noted that the GroupRATE framework may overly emphasize posterior dependence among features' coefficients. This undue focus could inadvertently inflate the model's sensitivity to feature removal and potentially understate the association strength with the response variable. To effectively address this concern, the authors must present robust evidence, including quantitative evaluations from real-world experiments. These evaluations should specifically assess the sensitivity of the method to feature removal.


**Audience:**

Yes, some TMLR's audience would be interested in knowing the findings of this paper. The paper tries to address the crucial issue of interpretability in neural networks, which is a significant concern in the machine-learning community. The proposed 'GroupRATE' criterion
provides a potential solution to identifying the group-level significance of features, which can be an important aspect of model interpretation. However, for this interest to be fully realized, the authors must adequately address the concerns raised by the reviewers about the method's general applicability and the comprehensiveness of its validation.

**Claims And Evidence:**

The authors have provided evidence for the utility of their GroupRATE framework through various tests conducted on both simulated and real-world data. However, both Reviewer 3a83 and Reviewer HwXW express justified concerns regarding the framework's broad applicability.

The authors' current application of their method is largely confined to a particular type of Bayesian network structure and datasets associated primarily with genomics and medical imaging applications. This limitation in the experimental design does not sufficiently substantiate their claim about the broad generality of the proposed method.

Furthermore, a major theoretical concern is raised by Reviewer HwXW. The proposed method may overly emphasize the posterior dependence amongst the features' coefficients, potentially diverting attention from the association strength with the response. This skewed focus could lead to an increased sensitivity of the model's performance to the removal of any feature.
To adequately address this issue, the authors are required to provide quantitative evaluations that delve into the sensitivity of their method to feature removal. Such evaluations would contribute to a more comprehensive understanding of the method's strengths and potential weaknesses. Currently, the provided evidence may not fully support the broad claims made in the paper concerning the overall applicability of their method.


**Resubmission Of Major Revision:**

The authors may consider submitting a major revision at a later time.